# Fine-to-Coarse Fairness-Informed Multi-View Clustering

**Shengju Yu** [1]  **Suyuan Liu** [2]  **Wenhao Shao** [3]  **Siwei Wang** [4]  **Dayu Hu** [5]  **Yiu-ming Cheung** [1]

## Abstract

In multi-view clustering (MVC), conventional anchor learning based models implicitly assume a uniform distribution of anchors across clusters, which could lead to inferior representation, especially when clusters vary significantly in size, as larger clusters require more anchors so as to adequately capture their intrinsic structural complexity. To alleviate this, we design a method termed FCFMVC that explicitly encourages proportional anchor allocation. To be specific, we transfer anchor allocation to discrete sample-cluster learning via bipartite graph bridge, and then backpropagate cluster state consisting of size and dispersion degree to guide anchor assignment. This allows the model to integrate cluster cardinality awareness and structural compactness directly into anchor distribution. On the other hand, we regard anchors as pseudo-samples, introduce an anchor-cluster indicator matrix on each view, and directly constrain the number of anchors assigned to each cluster within a tolerance margin. These two paths are further coupled through anchor-sample label alignment, and collaboratively facilitate anchor generation from fine-grained (anchor-level) to coarse-grained (cluster-level) structures. Besides, the entire optimization operation with linear time and space cost makes FCFMVC well-scalable to large-scale tasks. Experiments on datasets with diverse scales confirm the effectiveness of our FCFMVC.

## 1. Introduction

Multi-view clustering (MVC) has been widely recognized as an effective paradigm for exploiting the complementarity across diverse feature spaces to enhance clustering performance (Guo et al., 2024; Gu et al., 2024; Yu et al., 2024a; Xu et al., 2024). Among various MVC approaches, anchor learning based methods are increasingly prevalent owing to their robustness and ability to handle high-dimensional data (Qin et al., 2025; Shu et al., 2023; Chen et al., 2025a), and are employed in multiple fields, such as personalized advertising, social media discovery, etc (Li et al., 2024b; He et al., 2023; Lu & Feng, 2023). These methods typically construct the bipartite graph between samples and anchors to depict pairwise similarity, where anchors serve as representative prototypes that capture the underlying data structure (Li et al., 2023b; Yu et al., 2024b; Cui et al., 2023).

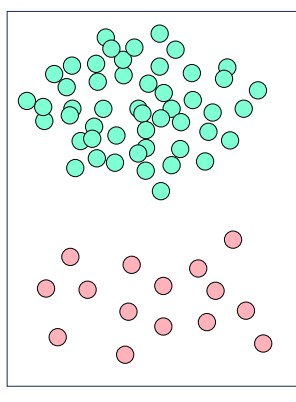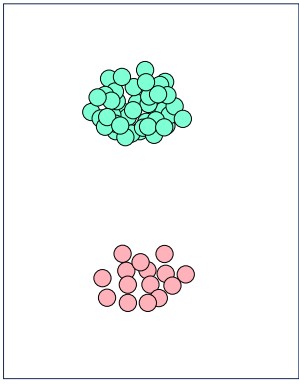

*(a)* Large Clusters  *(b)* Small Clusters

*Figure 1.* Illustration of Large Clusters and Small Clusters. The green and pink clusters contain 50 and 15 points, respectively. Large cluster: large size and low compactness (i.e., green cluster in (a)), or small size and high dispersion (pink cluster in (a)). Small cluster: Large size and high compactness (green cluster in (b)), or small size and low dispersion (pink cluster in (b)).

Despite notable success, a critical limitation is that they implicitly assume a uniform distribution of anchors across clusters, which could lead to insufficient detail extraction, since it overlooks cluster intrinsic characteristics. A larger cluster (as illustrated in Fig. 1), with its greater internal diversity and structural complexity, naturally requires more anchors so as to adequately capture its characteristics. Conversely, a compact, small cluster may be over-provisioned with anchors, introducing redundant computations and weakening the focus on meaningful patterns. This uniform anchor allocation mechanism not only degrades the quality of cluster

---

[1]Department of Computer Science, Hong Kong Baptist University, China. [2]National University of Defense Technology, China. [3]Guangxi University, China. [4]Intelligent Game and Decision Lab, China. [5]Northeastern University, China. yushengju@foxmail.com. Correspondence to: Yiu-ming Cheung <ymc@comp.hkbu.edu.hk>.

*Proceedings of the 43$^{rd}$ International Conference on Machine Learning*, Seoul, South Korea. PMLR 306, 2026. Copyright 2026 by the author(s).

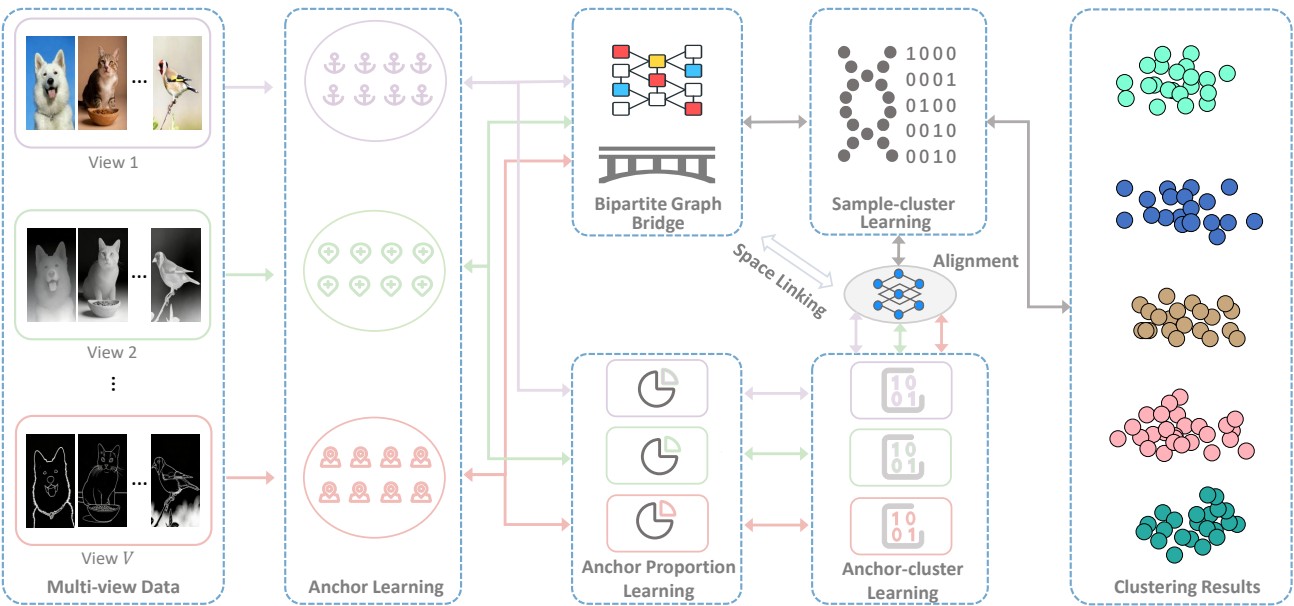

*Figure 2.* Framework of FCFMVC. It transfers anchor allocation to sample-cluster learning via bipartite graph bridge, and thereby reflects cluster state into anchor learning. Beyond that, it introduces anchor-cluster learning, and dynamically regulates the number of anchors per cluster per view via proportion constraint learning. The bridge further links anchor-clusters and sample-clusters, guaranteeing anchor-sample label alignment. Consequently, these two pathways are coupled together to collaboratively guide anchor assignment.

representation but also undermines the fairness of clustering outcomes, as larger clusters are systematically disadvantaged in capturing their intrinsic complexity.

To alleviate this issue, we give a fine-to-coarse fairness-informed MVC method, FCFMVC, to explicitly encourage anchor allocation to adapt to cluster state. Its overall framework is presented in Fig. 2. Specifically, we design a discrete optimization pathway that transfers anchor distribution supervision to the sample-cluster assignment via bipartite graph bridge, and then backpropagates cluster state consisting of size and intrinsic structural variation to guide anchor assignment. This enables the model to be aware of both cluster cardinality and intra-cluster compactness during anchor learning. On the other hand, we treat anchors as pseudo-samples and introduce per-view anchor-cluster indicator matrix. Through explicit constraints with tolerance margins, we directly regulate the proportion of anchors assigned to each cluster on each view. Then, we formulate a unified objective that jointly optimizes a consistency loss linking sample-clusters and anchor-clusters, thereby guaranteeing anchor-sample label alignment across views. Accordingly, these two pathways are collaboratively coupled, and create a feedback loop to guide anchor assignment from fine-grained (anchor-level) to coarse-grained (cluster-level) structures. Subsequently, the entire optimization pipeline is designed to operate with linear time and space complexity, which ensures the scalability of FCFMVC to large-scale scenes. To validate the effectiveness, we organize experiments on datasets with diverse scales. Results show that our FCFMVC delivers competitive clustering performance. Briefly, the contributions in this work include

1. We propose a multi-view clustering method named FCFMVC that introduces a dual-path cluster composition-aware mechanism to facilitate anchor allocation.

2. We develop an optimization pipeline with linear cost, which makes FCFMVC well-scalable to large-scale scenes.

3. We conduct extensive experiments on datasets spanning multiple scales to illustrate the effectiveness of FCFMVC.

## 2. Related Work

Multi-view clustering is a widely adopted paradigm for scalable and efficient data representation, typically by constructing one or more bipartite graph(s) connecting samples and anchors (Zhao et al., 2024b; Fu et al., 2025; He et al., 2025; Ji & Feng, 2025; Du et al., 2024). Recent advances in this area have focused on improving anchor selection, representation, and assignment to enhance clustering performance. For instance, Peng et al. (2025) integrate prior semantic coherence across views into anchor learning to improve its discriminability, while Li et al. (2023c) rearrange anchors through cross-view graph matching to boost its robustness against incomplete components. Feng et al. (2025) further extend anchor representation into a tensor space to capture high-order correlations. Unlike them, Zhao et al. (2024a) embed feature selection directly into anchor graph to refine feature set for more efficient graph factorization. Wang et al. (2025a) leverage attention mechanisms

to dynamically merge anchor graphs so as to focus on the most relevant information. Lao et al. (2024) jointly learn a diverse set of bipartite graphs to extract exclusive and consensus view structures, which enhances the flexibility of anchor-based representations. Moreover, Du et al. (2025) design an anchor-based deep unfolding network to iteratively refine cluster assignment. Long et al. (2025) integrate a low-frequency operator into predefined anchor graphs constructed by kernels to exploit intra-view sample smoothness. Li et al. (2024a) introduce tensorized label learning on anchor graphs, reducing label space complexity while preserving distinctive information. Qin et al. (2024) develop a consensus anchor learning to accelerate optimization. Despite achieving notable advances from various perspectives, these methods usually operate under uniform anchor allocation, which may lead to insufficient representation for larger clusters and undermine fairness in final clustering outcomes.

## 3. Methodology

Anchor learning based MVC can be basically formulated as

$$
\min_{\mathbf{A}_v, \mathbf{Z}_v, \mathbf{Z}} \sum_{v=1}^{V} \|\mathbf{X}_v - \mathbf{A}_v \mathbf{Z}_v\|_F^2 + \Phi(\mathbf{Z}_v, \mathbf{Z}) \tag{1}
$$
$$
\text{s.t. } \mathbf{A}_v^\top \mathbf{A}_v = \mathbf{I}_m, \mathbf{Z}_v \geq 0, \mathbf{Z}_v^\top \mathbf{1}_m = \mathbf{1}_n,
$$

where $\mathbf{X}_v \in \mathbb{R}^{d_v \times n}$, $\mathbf{A}_v \in \mathbb{R}^{d_v \times m}$ and $\mathbf{Z}_v \in \mathbb{R}^{m \times n}$ denote the data matrix, anchor matrix and bipartite graph on view $v$, respectively. $d_v$, $m$ and $n$ denote the data dimension, the number of anchors and samples, respectively. $\Phi$ denotes certain fusion strategies, and aims to merge all $\mathbf{Z}_v$ as a consensus graph $\mathbf{Z} \in \mathbb{R}^{m \times n}$ to aggregate view information. Then, the clustering results can be obtained by performing spectral partitioning on $\mathbf{Z}$.

To achieve more rational anchor allocation, we attempt to reflect cluster state into the anchor learning process. Considering that clustering results are derived from the consensus graph $\mathbf{Z}$, which indicates that $\mathbf{Z}$ encodes cluster-aware information, we skip the fusion stage, and choose to directly learn $\mathbf{Z}$ via anchors. Accordingly, we devise the loss as

$$
\min_{\mathbf{A}_v, \mathbf{Z}} \mathcal{L}_0(\mathbf{A}_v, \mathbf{Z}) = \sum_{v=1}^{V} \|\mathbf{X}_v - \mathbf{A}_v \mathbf{Z}\|_F^2 \tag{2}
$$
$$
\text{s.t. } \mathbf{A}_v^\top \mathbf{A}_v = \mathbf{I}_m, \mathbf{Z} \geq 0, \mathbf{Z}^\top \mathbf{1}_m = \mathbf{1}_n.
$$

Then, with the sample-sample similarity $\mathbf{Z}^\top \mathbf{Z}$, we introduce a sample-cluster indicator matrix $\mathbf{F} \in \mathbb{R}^{n \times k}$ where $k$ denotes the number of clusters to directly generate cluster labels. Consequently, we devise the loss as

$$
\max_{\mathbf{Z}, \mathbf{F}} \mathcal{L}_1(\mathbf{Z}, \mathbf{F}) = \mathrm{Tr}(\mathbf{F}^\top \mathbf{Z}^\top \mathbf{Z} \mathbf{F}) \tag{3}
$$
$$
\text{s.t. } \mathbf{Z} \geq 0, \mathbf{Z}^\top \mathbf{1}_m = \mathbf{1}_n, \mathbf{F} \in \{0,1\}^{n \times k}, \mathbf{F} \mathbf{1}_k = \mathbf{1}_n,
$$

where $\mathbf{F}$ plays a role in grouping $n$ samples into $k$ clusters.

Afterwards, we quantitatively characterize cluster state via two complementary metrics derived from $\mathbf{F}$. Especially, we have that $\mathbf{F}^\top \mathbf{F}$ is a diagonal matrix, where its $j$-th diagonal element corresponds to the number of samples in cluster $j$. Furthermore, to assess the distributional properties of clusters, we introduce the coefficient of variation to quantify the dispersion degree of each cluster relative to its central position, thereby providing a scale-invariant indicator of internal consistency. Therefore, we devise the loss as

$$
\min_{\mathbf{F}} \mathcal{L}_2(\mathbf{F}) = \left\| \frac{\mathbf{F}^\top \mathbf{F}}{n} - \frac{\mathbf{I}_k}{k} \right\|_F^2 + \frac{1}{k} \sum_{j=1}^{k} \left( \frac{\sigma_{\mathcal{C}_j}}{\mu_{\mathcal{C}_j}} - \frac{\bar{\sigma}}{\bar{\mu}} \right)^2
$$
$$
\text{s.t. } \mathbf{F} \in \{0,1\}^{n \times k}, \mathbf{F} \mathbf{1}_k = \mathbf{1}_n, \tag{4}
$$

where $\frac{\bar{\sigma}}{\bar{\mu}} = \frac{1}{k} \sum_{j=1}^{k} \frac{\sigma_{\mathcal{C}_j}}{\mu_{\mathcal{C}_j}}$, $\mu_{\mathcal{C}_j} = \frac{1}{n_{\mathcal{C}_j}} \sum_{i \in \mathcal{C}_j} d_{i,\mathcal{C}_j}$, $\sigma_{\mathcal{C}_j}^2 = \frac{1}{n_{\mathcal{C}_j}} \sum_{i \in \mathcal{C}_j} (d_{i,\mathcal{C}_j} - \mu_{\mathcal{C}_j})^2$, $d_{i,\mathcal{C}_j} = \|\mathbf{z}_i - \mathbf{c}_j\|_2$, $\mathbf{c}_j = \frac{1}{n_{\mathcal{C}_j}} \sum_{i \in \mathcal{C}_j} \mathbf{z}_i$. $\mathbf{z}_i$ denotes the $i$-th column of $\mathbf{Z}$. The notation $\mathcal{C}_j$ denotes cluster $j$. $n_{C_j}$ denotes the number of samples in cluster $j$, i.e., cluster size. $\mathbf{c}_j$ denotes the centroid of cluster $j$ in similarity space.

Serving as a bridge, $\mathbf{Z}$ enables the backpropagation of cluster-level characteristics containing cardinality and compactness into the learning of anchor matrix $\mathbf{A}_v$. This mechanism ensures that anchor representations are informed by the underlying cluster structure, thereby promoting consistency between anchor distribution and cluster properties.

On the other hand, anchors can be regarded as pseudo samples, and accordingly anchor-clusters should be consistent with sample-clusters. We introduce an anchor-cluster indicator matrix $\mathbf{G}_v \in \mathbb{R}^{m \times k}$ on each view to group anchors. Subsequently, we can transform $\mathbf{G}_v$ into sample-cluster space via $\mathbf{Z}^\top \mathbf{G}_v \in \mathbb{R}^{n \times k}$ to align with $\mathbf{F}$. Thus, we devise the loss as

$$
\min_{\mathbf{Z}, \mathbf{F}, \mathbf{G}_v} \mathcal{L}_3(\mathbf{Z}, \mathbf{F}, \mathbf{G}_v) = \sum_{v=1}^{V} \|\mathbf{F} - \mathbf{Z}^\top \mathbf{G}_v\|_F^2
$$
$$
\text{s.t. } \mathbf{Z} \geq 0, \mathbf{Z}^\top \mathbf{1}_m = \mathbf{1}_n, \mathbf{F} \in \{0,1\}^{n \times k}, \tag{5}
$$
$$
\mathbf{F} \mathbf{1}_k = \mathbf{1}_n, \mathbf{G}_v \in \{0,1\}^{m \times k}, \mathbf{G}_v \mathbf{1}_k = \mathbf{1}_m,
$$

where $\mathbf{G}_v$ partitions $m$ anchors on view $v$ into $k$ clusters.

Further, since each column sum of $\mathbf{G}_v$ reflects the number of anchors assigned to the corresponding cluster, we can bound its column sum within a tolerance margin. This enables adaptive fluctuation in anchor allocation, allowing the distribution of anchors to respond to cluster characteristics. Hence, we devise the loss as

$$
\min_{\mathbf{G}_v} \mathcal{L}_4(\mathbf{G}_v) = \sum_{v=1}^{V} \sum_{j=1}^{k} \left[ \max\left( 0, \frac{\mathbf{1}_m^\top [\mathbf{G}_v]_{.,j}}{m} - \frac{1+\delta}{k} \right)^2 \right.
$$

$$+ \max\left(0, \frac{1-\delta}{k} - \frac{\mathbf{1}_m^\top [\mathbf{G}_v]_{.,j}}{m}\right)^2\Bigg]$$

$$\text{s.t. } \mathbf{G}_v \in \{0,1\}^{m \times k}, \mathbf{G}_v \mathbf{1}_k = \mathbf{1}_m, \quad (6)$$

where $0 < \delta < 1$ is a predefined constant.

Integrating the above, we devise the final loss function as

$$\min_{\mathbf{A}_v, \mathbf{Z}, \mathbf{F}, \mathbf{G}_v} \mathcal{L}_0 - \lambda_1 \mathcal{L}_1 + \lambda_2 \mathcal{L}_2 + \lambda_3 \mathcal{L}_3 + \lambda_4 \mathcal{L}_4$$
$$\text{s.t. } \mathbf{A}_v^\top \mathbf{A}_v = \mathbf{I}_m, \mathbf{Z} \geq 0, \mathbf{Z}^\top \mathbf{1}_m = \mathbf{1}_n, \mathbf{F}\mathbf{1}_k = \mathbf{1}_n, \quad (7)$$
$$\mathbf{F} \in \{0,1\}^{n \times k}, \mathbf{G}_v \in \{0,1\}^{m \times k}, \mathbf{G}_v \mathbf{1}_k = \mathbf{1}_m.$$

## 4. Optimization

We solve the multi-variable task (7) via alternating updates.

$\mathbf{A}_v$ **Sub-problem:** With $\mathbf{Z}$, $\mathbf{G}_v$ and $\mathbf{F}$ fixed, the problem (7) reduces to

$$\min_{\mathbf{A}_v^\top \mathbf{A}_v = \mathbf{I}_m} \mathcal{L}_0. \quad (8)$$

Owing to view independence, we can transform (8) into

$$\min_{\mathbf{A}_v^\top \mathbf{A}_v = \mathbf{I}_m} \|\mathbf{X}_v - \mathbf{A}_v \mathbf{Z}\|_F^2. \quad (9)$$

Expanding $F$-norm via trace operation, we further have

$$\max_{\mathbf{A}_v^\top \mathbf{A}_v = \mathbf{I}_m} \text{Tr}\left(\mathbf{X}_v \mathbf{Z}^\top \mathbf{A}_v^\top\right). \quad (10)$$

The optimal $\mathbf{A}_v$ is $\mathbf{U}\mathbf{V}^\top$. $\mathbf{U}$ and $\mathbf{V}$ are the SVD of $\mathbf{X}_v \mathbf{Z}^\top$.

$\mathbf{Z}$ **Sub-problem:** With $\mathbf{A}_v$, $\mathbf{G}_v$ and $\mathbf{F}$ fixed, the problem (7) reduces to

$$\min_{\mathbf{Z} \geq 0, \mathbf{Z}^\top \mathbf{1}_m = \mathbf{1}_n} \mathcal{L}_0 - \lambda_1 \mathcal{L}_1 + \lambda_3 \mathcal{L}_3. \quad (11)$$

We then reformulate it into the following trace form:

$$\min_{\mathbf{Z}} \sum_{v=1}^{V} \text{Tr}\left(\mathbf{Z}^\top \mathbf{Z} - 2\mathbf{X}_v^\top \mathbf{A}_v \mathbf{Z} - 2\lambda_3 \mathbf{F}\mathbf{G}_v^\top \mathbf{Z}\right)$$
$$-\lambda_1 \text{Tr}\left(\mathbf{F}^\top \mathbf{Z}^\top \mathbf{Z}\mathbf{F}\right) + \lambda_3 \sum_{v=1}^{V} \text{Tr}\left(\mathbf{Z}^\top \mathbf{G}_v \mathbf{G}_v^\top \mathbf{Z}\right) \quad (12)$$
$$\text{s.t. } \mathbf{Z} \geq 0, \mathbf{Z}^\top \mathbf{1}_m = \mathbf{1}_n.$$

To address this problem, we first establish **Theorem 1**.

**Theorem 1.** *For any $\mathbf{F} \in \{0,1\}^{n \times k}$ satisfying $\mathbf{F}\mathbf{1}_k = \mathbf{1}_n$, we have* $\text{Tr}\left(\mathbf{F}^\top \mathbf{S}\mathbf{F}\right) = \sum_{i=1}^{k} \sum_{a \in \mathcal{T}_i} \sum_{b \in \mathcal{T}_i} S_{ab}$, *where $\mathcal{T}_i \subseteq \{1, 2, \cdots, n\}$ denotes the set of row indices corresponding to the non-zero entries in the $i$-th column of $\mathbf{F}$.*

Building upon the results in **Theorem 1**, we further derive the following theorem:

**Theorem 2.** *For $\mathbf{Z} > \mathbf{0}_{m \times n}$ satisfying $\mathbf{Z}^\top \mathbf{1}_m = \mathbf{1}_n$, we have that maximizing $\text{Tr}\left(\mathbf{F}^\top \mathbf{Z}^\top \mathbf{Z}\mathbf{F}\right)$ on $\mathbf{Z}$ is equivalent to*

$$\max_{\mathbf{Z}_{:,j}} \mathbf{Z}_{:,j}^\top \left(\mathbf{Z}_{:,j} + 2 \sum_{a \in \mathcal{T}_i \setminus \{j\}} \mathbf{Z}_{:,a}\right), \forall j \in \mathcal{T}_i, \quad (13)$$

*where $i \in \{1, 2, \cdots, k\}$ and $\bigcup_{i=1}^{k} \mathcal{T}_i = \{1, 2, \cdots, n\}$.*

**Remark 1.** *Each sub-problem involves only one column $\mathbf{Z}_{:,j}$ and depends linearly on the remaining columns in the same block $\mathcal{T}_i$, rendering it well-suited for alternating updates.*

Given **Theorem 2** and the column-wise nature of the constraints on $\mathbf{Z}$, we can update $\mathbf{Z}$ column by column. Consequently, we convert (12) to the following equivalent form:

$$\min_{\mathbf{Z}_{:,j}} \frac{1}{2} \mathbf{Z}_{:,j}^\top \left((V - \lambda_1)\mathbf{I}_m + \lambda_3 \sum_{v=1}^{V} \mathbf{G}_v \mathbf{G}_v^\top\right) \mathbf{Z}_{:,j}$$
$$- \left(\mathbf{D}_{j,:} + \lambda_1 \sum_{a \in \mathcal{C}_i \setminus \{j\}} \mathbf{Z}_{:,a}^\top + \lambda_3 \mathbf{H}_{j,:}\right) \mathbf{Z}_{:,j} \quad (14)$$
$$\text{s.t. } \mathbf{Z}_{:,j} \geq 0, \mathbf{Z}_{:,j}^\top \mathbf{1}_m = 1,$$

where $\mathbf{D} = \sum_{v=1}^{V} \mathbf{X}_v^\top \mathbf{A}_v$, $\mathbf{H} = \mathbf{F} \sum_{v=1}^{V} \mathbf{G}_v^\top$.

Before solving this problem, we give the following theorem:

**Theorem 3.** *For $\mathbf{G}_v \in \{0,1\}^{m \times k}$ satisfying $\mathbf{G}_v \mathbf{1}_k = \mathbf{1}_m$, if either the scalar $a \neq 0$ or $b \neq 0$, then we have that the matrix $a\mathbf{I}_m + b\sum_{v=1}^{V} \mathbf{G}_v^\top \mathbf{G}_v$ is non-zero.*

Note that the regularization hyper-parameters are typically non-zero. With **Theorem 3**, we have that (14) is a quadratic programming problem, and accordingly can be solved using off-the-shelf software packages.

**Remark 2.** *Even if $\lambda_3$ is set to 0 and the term $(V - \lambda_1)\mathbf{I}_m + \lambda_3 \sum_{v=1}^{V} \mathbf{G}_v \mathbf{G}_v^\top$ reduces to a zero matrix, (14) can still be readily solved, as it simplifies to a standard linear programming problem.*

**Remark 3.** *Instead of directly constructing the term $\sum_{a \in \mathcal{C}_i \setminus \{j\}} \mathbf{Z}_{:,a}^\top$ for each $\mathbf{Z}_{:,j}$, we first pre-compute a common summation $\sum_{a \in \mathcal{C}_i} \mathbf{Z}_{:,a}^\top$ for all $\mathbf{Z}_{:,j}$ in the same cluster, and then subtract current $\mathbf{Z}_{:,j}$ from it, which effectively avoids repeated summations across the cluster members. Following the update of $\mathbf{Z}_{:,j}$, the pre-commuted summation is efficiently revised by adding the difference between the new and previous values of $\mathbf{Z}_{:,j}$ for next $\mathbf{Z}_{:,j}$ optimization.*

$\mathbf{G}_v$ **Sub-problem:** With $\mathbf{A}_v$, $\mathbf{Z}$ and $\mathbf{F}$ fixed, the problem (7) reduces to

$$\min_{\mathbf{G}_v \in \{0,1\}^{m \times k}, \mathbf{G}_v \mathbf{1}_k = \mathbf{1}_m} \lambda_3 \mathcal{L}_3 + \lambda_4 \mathcal{L}_4. \quad (15)$$

With view independence, we equivalently convert (15) into

$$\min_{\mathbf{G}_v} \lambda_3 \left\| \mathbf{F} - \mathbf{Z}^\top \mathbf{G}_v \right\|_F^2 + \lambda_4 \sum_{i=1}^{k} f \left( \frac{\mathbf{1}_m^\top [\mathbf{G}_v]_{:,i}}{m} \right) \quad (16)$$

$$\text{s.t. } \mathbf{G}_v \in \{0,1\}^{m \times k}, \mathbf{G}_v \mathbf{1}_k = \mathbf{1}_m,$$

where $f(x) = \max \left( 0, x - \frac{1+\delta}{k} \right)^2 + \max \left( 0, \frac{1-\delta}{k} - x \right)^2$.

Given the discreteness of feasible regions, we can optimize (15) via row-wise scanning using binary learning. Specially, for the $j$-th row of $\mathbf{G}_v$ (i.e., anchor $j$), we pick out the case where the loss decrease is the most significant, and then set the value at this position to 1 while setting other elements in this row to 0. That is, for anchor $j$, we assign it to the cluster that yields the most substantial reduction in the loss.

To quantify the loss change, we have the following theorem:

**Theorem 4.** *For any anchor $j \in \{1, 2, \cdots, m\}$ on view $v \in \{1, 2, \cdots, V\}$, when reassigning it from cluster $c$ to cluster $l$, the change in $\left\| \mathbf{F} - \mathbf{Z}^\top \mathbf{G}_v \right\|_F^2$ is equal to*

$$\Delta_{zgf} := 2\mathbf{Z}_{j,:} \left( \mathbf{Z}_{j,:}^\top + \mathbf{R}_{:,c} - \mathbf{R}_{:,l} \right), \quad (17)$$

*where $\mathbf{R} = \mathbf{F} - \mathbf{Z}^\top \mathbf{G}_v$.*

Note that for any $q \notin \{c, l\}$, the cluster components maintain unchanged. Therefore, we have that the change in $\sum_{i=1}^{k} f \left( \frac{\mathbf{1}_m^\top [\mathbf{G}_v]_{:,i}}{m} \right)$ is equal to

$$\Delta_{gvm} := f \left( h_c - \frac{1}{m} \right) - f(h_c) + f \left( h_l + \frac{1}{m} \right) - f(h_l), \quad (18)$$

where $h_c = \frac{1}{m} \sum_{i=1}^{m} [\mathbf{G}_v]_{ic}, h_l = \frac{1}{m} \sum_{i=1}^{m} [\mathbf{G}_v]_{il}$.

With **Theorem 4** and (18), thus, when reassigning anchor $j$ from cluster $c$ to cluster $l$, the loss change in (16) is

$$\Delta_{zg} := \lambda_3 \Delta_{zgf} + \lambda_4 \Delta_{gvm}. \quad (19)$$

Then, we traverse $l \in \{1, 2, \cdots, k\}$ to find $l^*$ corresponding to the maximal loss reduction by

$$l^* = \arg \min_l \Delta_{zg}, \quad (20)$$

and accordingly update $[\mathbf{G}_v]_{jc} = 0$ and $[\mathbf{G}_v]_{jl^*} = 1$.

**Remark 4.** *A positive value of $\Delta_{zg}$ indicates loss increase while a negative value indicates loss decrease. Therefore, in (20), the operation is to take $\arg \min$ rather than $\arg \max$.*

**Remark 5.** *Once one row scanning is completed, it needs to update $\mathbf{R}$ and the cluster size proportions $h_c$ and $h_{l^*}$ in preparation for the next row scanning, as $\mathbf{G}_v$ has been modified. Specifically, the update rules are given by $\mathbf{R}_{:,l^*} \leftarrow \mathbf{R}_{:,l^*} - \mathbf{Z}_{j,:}^\top$ and $\mathbf{R}_{:,c} \leftarrow \mathbf{R}_{:,c} + \mathbf{Z}_{j,:}^\top; h_{l^*} \leftarrow h_{l^*} + \frac{1}{m}$ and $h_c \leftarrow h_c - \frac{1}{m}$.*

---

**Algorithm 1** Pipeline of Updating $\mathbf{G}_v$

**Input**: $\mathbf{F}, \mathbf{Z}, \mathbf{G}_v$
**Output**: Updated $\mathbf{G}_v$

1: **for** $j = 1$ to $m$ **do**
2:    $c \leftarrow$ the index such that $[\mathbf{G}_v]_{j,c} = 1$
3:    **for** $l = 1$ to $k$ **do**
4:       Compute the loss change $\Delta_{zg}$ via (19)
5:    **end for**
6:    Set self-assignment loss to zero, $\Delta_{zg}(c) \leftarrow 0$
7:    Find $l^* = \arg \min_{l \in \{1, \ldots, k\}} \Delta_{zg}$
8:    Update $[\mathbf{G}_v]_{jc} : 1 \mapsto 0, [\mathbf{G}_v]_{jl^*} : 0 \mapsto 1$
9:    Update $\mathbf{R}_{:,l^*} \leftarrow \mathbf{R}_{:,l^*} - \mathbf{Z}_{j,:}^\top, \mathbf{R}_{:,c} \leftarrow \mathbf{R}_{:,c} + \mathbf{Z}_{j,:}^\top$
    Update $h_{l^*} \leftarrow h_{l^*} + \frac{1}{m}, h_c \leftarrow h_c - \frac{1}{m}$
10: **end for**

---

Algorithm 1 outlines the pipeline of updating $\mathbf{G}_v$.

**$\mathbf{F}$ Sub-problem:** With $\mathbf{A}_v, \mathbf{Z}$ and $\mathbf{G}_v$ fixed, the problem (7) reduces to

$$\min_{\mathbf{F} \in \{0,1\}^{n \times k}, \mathbf{F}\mathbf{1}_k = \mathbf{1}_n} -\lambda_1 \mathcal{L}_1 + \lambda_2 \mathcal{L}_2 + \lambda_3 \mathcal{L}_3. \quad (21)$$

Inspired by the strategies tackling (16), we address this problem by reallocating each sample $j \in \{1, 2, \cdots, n\}$ from cluster $r$ to cluster $t$ to identify the maximal loss reduction.

With **Theorem 1**, we can derive that the change in $\mathcal{L}_1$ equals

$$\Delta_{fz} := 2\mathbf{Z}_{:,j}^\top \left( \sum_{a \in \mathcal{C}_t} \mathbf{Z}_{:,a} - \sum_{a \in \mathcal{C}_r, a \neq j} \mathbf{Z}_{:,a} \right). \quad (22)$$

By virtue of trace transformation and element-wise expansion, we have that the change in $\mathcal{L}_3$ equals

$$\Delta_{fzg} := 2 \left[ \mathbf{Z}^\top \sum_{v=1}^{V} \mathbf{G}_v \right]_{jr} - 2 \left[ \mathbf{Z}^\top \sum_{v=1}^{V} \mathbf{G}_v \right]_{jt}. \quad (23)$$

Then, for the term $\left\| \frac{\mathbf{F}^\top \mathbf{F}}{n} - \frac{\mathbf{I}_k}{k} \right\|_F^2$ in $\mathcal{L}_2$, its change equals

$$\Delta_{si} := \frac{2 \left( n_{\mathcal{C}_t} - n_{\mathcal{C}_r} + 1 \right)}{n^2}. \quad (24)$$

For the term $\frac{1}{k} \sum_{i=1}^{k} \left( \frac{\sigma_{c_i}}{\mu_{c_i}} - \frac{\bar{\sigma}}{\bar{\mu}} \right)^2$ in $\mathcal{L}_2$, we have the following theorem:

**Theorem 5.** *For any sample $j \in \{1, 2, \cdots, n\}$, when reallocating it from cluster $r$ to cluster $t$, the change in $\frac{1}{k} \sum_{i=1}^{k} \left( \frac{\sigma_{c_i}}{\mu_{c_i}} - \frac{\bar{\sigma}}{\bar{\mu}} \right)^2$ is equal to*

$$\Delta_{cv} := \frac{(k-1) \left( \widehat{\Delta}_r^2 + \widehat{\Delta}_t^2 \right) - 2\widehat{\Delta}_r \widehat{\Delta}_t - 2k \left( d_r \widehat{\Delta}_r + d_t \widehat{\Delta}_t \right)}{k^2}, \quad (25)$$

**Algorithm 2** Pipeline of Updating $\mathbf{F}$

**Input**: $\mathbf{F}, \mathbf{Z}, \mathbf{G}_v$

**Output**: Updated $\mathbf{F}$

1: **for** $r = 1$ to $k$ **do**
2: $\quad \widehat{\mathcal{C}}_r = \mathcal{C}_r$
3: $\quad$ **for** $q = 1$ to the size of $\widehat{\mathcal{C}}_r$ **do**
4: $\qquad j = \widehat{\mathcal{C}}_r(q)$
5: $\qquad$ Compute $\frac{\sigma'_{\mathcal{C}_r}}{\mu'_{\mathcal{C}_r}}$ via (26)
6: $\qquad$ **for** $t = 1$ to $k$ **do**
7: $\qquad\quad$ Compute $\frac{\sigma'_{\mathcal{C}_t}}{\mu'_{\mathcal{C}_t}}$ via (26)
8: $\qquad\quad$ Compute $\Delta_{tf}$ via (27)
9: $\qquad$ **end for**
10: $\qquad \Delta_{tf}(r) \leftarrow 0$
11: $\qquad t^* = \arg\min_{t \in \{1,\dots,k\}} \Delta_{tf}$
12: $\qquad$ Update $\mathbf{F}_{jr} : 1 \mapsto 0, \ \mathbf{F}_{jt^*} : 0 \mapsto 1$
13: $\qquad$ Update $n_{\mathcal{C}_r} \leftarrow n_{\mathcal{C}_r} - 1, \ n_{\mathcal{C}_{t^*}} \leftarrow n_{\mathcal{C}_{t^*}} + 1$
14: $\qquad$ Update $\frac{\bar{\sigma}}{\bar{\mu}} \leftarrow \frac{\bar{\sigma}}{\bar{\mu}} + \frac{1}{k}\left(\widehat{\Delta}_r + \widehat{\Delta}_{t^*}\right)$
15: $\qquad$ Update $\frac{\sigma_{\mathcal{C}_r}}{\mu_{\mathcal{C}_r}} \leftarrow \frac{\sigma'_{\mathcal{C}_r}}{\mu'_{\mathcal{C}_r}}, \ \frac{\sigma_{\mathcal{C}_{t^*}}}{\mu_{\mathcal{C}_{t^*}}} \leftarrow \frac{\sigma'_{\mathcal{C}_{t^*}}}{\mu'_{\mathcal{C}_{t^*}}}$
16: $\qquad$ Update cluster members $\mathcal{C}_r, \mathcal{C}_{t^*}$
17: $\qquad$ Update $S_{D1}(r) \leftarrow S_{D1}(r) - d_{j,\mathcal{C}_r}$
18: $\qquad$ Update $S_{D2}(r) \leftarrow S_{D2}(r) - d^2_{j,\mathcal{C}_r}$,
19: $\qquad$ Update $S_{D1}(t^*) \leftarrow S_{D1}(t^*) + d_{j,\mathcal{C}_{t^*}}$,
20: $\qquad$ Update $S_{D2}(t^*) \leftarrow S_{D2}(t^*) + d^2_{j,\mathcal{C}_{t^*}}$,
21: $\qquad$ Update $S_C(r) \leftarrow S_C(r) - \mathbf{Z}_{:,j}$,
22: $\qquad$ Update $S_C(t^*) \leftarrow S_C(t^*) + \mathbf{Z}_{:,j}$.
23: $\quad$ **end for**
24: **end for**
25: Update $\mathbf{c}_i, \ d_{j,\mathcal{C}_i}, \ \forall i \in \{\text{altered clusters}\}$
26: Compute $\frac{\sigma_{\mathcal{C}_i}}{\mu_{\mathcal{C}_i}}, \ \forall i \in \{\text{altered clusters}\}, \ \frac{\bar{\sigma}}{\bar{\mu}}$

where $d_r = \frac{\bar{\sigma}}{\bar{\mu}} - \frac{\sigma_{\mathcal{C}_r}}{\mu_{\mathcal{C}_r}}$, $d_t = \frac{\bar{\sigma}}{\bar{\mu}} - \frac{\sigma_{\mathcal{C}_t}}{\mu_{\mathcal{C}_t}}$, $\widehat{\Delta}_r = \frac{\sigma'_{\mathcal{C}_r}}{\mu'_{\mathcal{C}_r}} - \frac{\sigma_{\mathcal{C}_r}}{\mu_{\mathcal{C}_r}}$ and $\widehat{\Delta}_t = \frac{\sigma'_{\mathcal{C}_t}}{\mu'_{\mathcal{C}_t}} - \frac{\sigma_{\mathcal{C}_t}}{\mu_{\mathcal{C}_t}}$. $\frac{\sigma'_{\mathcal{C}_r}}{\mu'_{\mathcal{C}_r}}$ denotes the cluster state after sample reallocating while $\frac{\sigma_{\mathcal{C}_r}}{\mu_{\mathcal{C}_r}}$ denotes the state before sample reallocating. Likewise for $\frac{\sigma'_{\mathcal{C}_t}}{\mu'_{\mathcal{C}_t}}$ and $\frac{\sigma_{\mathcal{C}_t}}{\mu_{\mathcal{C}_t}}$.

**Remark 6.** *After reallocating sample $j$ from cluster $r$ to cluster $t$, the cluster components undergo changes. Accordingly, it becomes necessary to recalculate the cluster centroids $(\mathbf{c}_r, \mathbf{c}_t)$ and the intra-cluster similarity distances $(d_{i,\mathcal{C}_r}, d_{p,\mathcal{C}_t}, \forall i \in \mathcal{C}_r, \forall p \in \mathcal{C}_t)$. Updating all $d_{i,\mathcal{C}_r}$ and $d_{p,\mathcal{C}_t}$ requires $\mathcal{O}(n_{\mathcal{C}_r} + n_{\mathcal{C}_t})$ computational cost. That is, each reallocation incurs $\mathcal{O}(n_{\mathcal{C}_r} + n_{\mathcal{C}_t})$ cost. Therefore, scanning sample $j$ totally takes $\mathcal{O}(n)$ cost, and a complete round of sample scanning takes $\mathcal{O}(n^2)$ cost.*

**Remark 7.** *To alleviate the excessive computational burden, we leverage a block coordinate descent inspired strategy by treating the cluster centroids as fixed throughout the sample scanning phase. Only after finishing a complete round of*

*sample scanning, we recalculate all centroids and similarity distances for the clusters with altered member compositions for the next scanning round. Accordingly, we can derive*

$$
\frac{\sigma'_{\mathcal{C}_r}}{\mu'_{\mathcal{C}_r}} = \frac{\sqrt{n'_{\mathcal{C}_r}\left(\sum_{i \in \mathcal{C}_r \setminus \{j\}} d^2_{i,\mathcal{C}_r}\right) - \left(\sum_{i \in \mathcal{C}_r \setminus \{j\}} d_{i,\mathcal{C}_r}\right)^2}}{\sum_{i \in \mathcal{C}_r \setminus \{j\}} d_{i,\mathcal{C}_r}}
$$

$$
\frac{\sigma'_{\mathcal{C}_t}}{\mu'_{\mathcal{C}_t}} = \frac{\sqrt{n'_{\mathcal{C}_t}\left(\sum_{i \in \mathcal{C}_t} d^2_{i,\mathcal{C}_t} + d^2_{j,\mathcal{C}_t}\right) - \left(\sum_{i \in \mathcal{C}_t} d_{i,\mathcal{C}_t} + d_{j,\mathcal{C}_t}\right)^2}}{\sum_{i \in \mathcal{C}_t} d_{i,\mathcal{C}_t} + d_{j,\mathcal{C}_t}},
$$

$$(26)$$

*where $n'_{\mathcal{C}_r} = n_{\mathcal{C}_r} - 1$ and $n'_{\mathcal{C}_t} = n_{\mathcal{C}_t} + 1$.*

Therefore, we can derive the change in loss (21) as

$$
\Delta_{tf} = -\lambda_1 \Delta_{fz} + \lambda_2 \left(\Delta_{si} + \Delta_{cv}\right) + \lambda_3 \Delta_{fzg}. \quad (27)
$$

With $t^* = \arg\min_t \Delta_{tf}$, we update $\mathbf{F}_{jr} = 0$ and $\mathbf{F}_{jt^*} = 1$.

**Remark 8.** *We can construct $\sum_{i \in \mathcal{C}_r} d^2_{i,\mathcal{C}_r}$ and $\sum_{i \in \mathcal{C}_r} d_{i,\mathcal{C}_r}$, and then subtract $d^2_{j,\mathcal{C}_r}$ and $d_{j,\mathcal{C}_r}$ from these two terms respectively to equivalently generate $\sum_{i \in \mathcal{C}_r \setminus \{j\}} d^2_{i,\mathcal{C}_r}$ and $\sum_{i \in \mathcal{C}_r \setminus \{j\}} d_{i,\mathcal{C}_r}$, which avoids redundant repeated summations for different $j$. With the updated summation set $\{\sum_{a \in \mathcal{C}_p} \mathbf{Z}_{:,a}\}^k_{p=1}$ established in the optimization of $\mathbf{Z}$, we can update $\mathbf{F}$ further through partition-wise optimization during the row-wise optimization process.*

**Algorithm 2** outlines the pipeline of updating $\mathbf{F}$, where $S_{D1}(r) = \sum_{i \in \mathcal{C}_r} d_{i,\mathcal{C}_r}$, $S_{D2}(r) = \sum_{i \in \mathcal{C}_r} d^2_{i,\mathcal{C}_r}$, $S_{D1}(t^*) = \sum_{i \in \mathcal{C}_{t^*}} d_{i,\mathcal{C}_{t^*}}$, $S_{D2}(t^*) = \sum_{i \in \mathcal{C}_{t^*}} d^2_{i,\mathcal{C}_{t^*}}$, $S_C(r) = \sum_{a \in \mathcal{C}_r} \mathbf{Z}_{:,a}$, and $S_C(t^*) = \sum_{a \in \mathcal{C}_{t^*}} \mathbf{Z}_{:,a}$.

At this stage, based on the derived update rules, we summarize the solution pipeline for (7), as presented in **Algorithm 3**, where $\mathcal{L}_{prev}$ and $\mathcal{L}_{curr}$ denote the loss values at the previous and current optimization steps, respectively.

---

**Algorithm 3** The proposed FCFMVC

**Input**: Multi-view data $\{\mathbf{X}_v\}^V_{v=1}$, number of clusters $k$, number of anchors $m$, parameters $\lambda_1, \lambda_2, \lambda_3, \lambda_4$

**Output**: Clustering labels $\mathbf{y} \in \{1, \dots, k\}^n$

**Initialize**: $\mathbf{A}_v, \mathbf{Z}, \mathbf{G}_v, \mathbf{F}$

1: **repeat**
2: $\quad$ Update $\mathbf{A}_v$ by (10)
3: $\quad$ Update $\mathbf{Z}$ by (14)
4: $\quad$ Update $\mathbf{G}_v$ by **Algorithm 1**
5: $\quad$ Update $\mathbf{F}$ by **Algorithm 2**
6: **until** $\left(\mathcal{L}_{prev} - \mathcal{L}_{curr}\right)/|\mathcal{L}_{prev}| \leq 1e - 5$
7: Extract clustering labels from $\mathbf{F}$: $y_i = j$ where $F_{ij} = 1$

---

Regarding its complexity, we have the following theorem:

**Theorem 6.** *The proposed method achieves linear time and space complexity with respect to the sample size, and accordingly is suitable for large-scale scenarios.*

*Table 1.* Clustering Result Comparison. Top-2 results are marked in red and blue, respectively.

| Dataset | MSGL | UOVSC | PFSC | SFMC | PGSC | ORTF | FMAC | LTBL | HOMGC | DGF | MFSA | UDBG | TTDM | Ours |
|---|---|---|---|---|---|---|---|---|---|---|---|---|---|---|
| | | | | | | ACC(%) | | | | | | | | |
| FORETYP | 75.53 | 84.51 | 66.99 | 50.67 | 37.05 | 31.93 | 78.37 | 65.67 | 73.62 | 69.42 | 71.32 | 78.39 | 80.42 | 82.79 |
| CITSER | 22.65 | 25.30 | 24.92 | 21.83 | 23.15 | 21.92 | 48.63 | 44.56 | 52.46 | 32.34 | 29.78 | 42.45 | 34.74 | 49.58 |
| NUSWINE | 28.60 | 13.46 | 12.74 | 28.28 | 37.05 | 15.91 | 11.70 | 27.43 | 26.52 | 31.52 | 19.43 | 28.13 | 25.45 | 38.62 |
| NUECTEN | 23.72 | 24.35 | 22.02 | 22.81 | 23.57 | 12.14 | 22.44 | 24.67 | 21.54 | 18.42 | 19.72 | 24.28 | 21.17 | 25.13 |
| YTFATEN | 76.29 | - | - | 55.80 | - | 15.09 | 72.93 | - | - | 67.72 | 71.11 | 63.86 | 68.93 | 78.29 |
| YTFSFLF | 22.78 | - | - | - | - | 14.46 | 23.98 | - | - | 19.42 | 20.24 | 26.80 | 17.32 | 26.32 |
| | | | | | | NMI(%) | | | | | | | | |
| FORETYP | 53.58 | 62.56 | 47.75 | 36.02 | 2.64 | 11.46 | 51.44 | 32.43 | 34.28 | 47.26 | 39.53 | 55.43 | 46.83 | 63.42 |
| CITSER | 2.40 | 3.93 | 4.53 | 6.49 | 1.59 | 5.34 | 27.07 | 6.37 | 5.72 | 7.33 | 8.32 | 18.38 | 17.83 | 24.93 |
| NUSWINE | 1.63 | 10.29 | 6.94 | 1.19 | 2.64 | 11.23 | 6.31 | 5.38 | 4.32 | 7.51 | 10.51 | 3.89 | 9.72 | 11.08 |
| NUECTEN | 3.11 | 10.93 | 1.90 | 1.54 | 2.55 | 8.08 | 11.02 | 7.23 | 10.32 | 11.16 | 6.62 | 10.07 | 8.74 | 11.12 |
| YTFATEN | 81.69 | - | - | 77.46 | - | 8.77 | 79.58 | - | - | 61.37 | 62.41 | 72.84 | 55.76 | 83.21 |
| YTFSFLF | 1.86 | - | - | - | - | 11.14 | 21.78 | - | - | 20.21 | 19.38 | 8.13 | 16.79 | 23.67 |
| | | | | | | PURITY(%) | | | | | | | | |
| FORETYP | 73.41 | 84.51 | 67.56 | 50.67 | 38.07 | 37.48 | 78.76 | 45.62 | 49.51 | 62.43 | 67.12 | 78.39 | 59.46 | 81.23 |
| CITSER | 22.80 | 26.84 | 26.12 | 24.67 | 23.29 | 21.98 | 52.00 | 34.82 | 31.63 | 42.31 | 39.21 | 42.60 | 47.26 | 45.87 |
| NUSWINE | 29.11 | 33.04 | 31.82 | 28.91 | 38.07 | 28.77 | 31.18 | 27.72 | 26.31 | 24.72 | 28.24 | 29.47 | 27.57 | 39.58 |
| NUECTEN | 24.38 | 31.27 | 24.01 | 22.92 | 23.85 | 23.08 | 33.25 | 26.49 | 24.52 | 29.71 | 34.32 | 30.57 | 28.46 | 34.16 |
| YTFATEN | 77.82 | - | - | 74.10 | - | 19.41 | 78.94 | - | - | 66.37 | 62.75 | 68.87 | 65.82 | 80.18 |
| YTFSFLF | 26.84 | - | - | - | - | 26.62 | 33.13 | - | - | 31.98 | 31.57 | 27.21 | 33.24 | 34.02 |
| | | | | | | FSCORE(%) | | | | | | | | |
| FORETYP | 61.18 | 72.55 | 61.89 | 52.08 | 43.18 | 27.09 | 63.97 | 45.57 | 48.26 | 39.73 | 36.12 | 64.73 | 33.67 | 72.17 |
| CITSER | 29.97 | 25.41 | 25.84 | 29.82 | 29.34 | 28.21 | 36.70 | 37.36 | 38.92 | 31.12 | 35.87 | 31.10 | 29.78 | 40.24 |
| NUSWINE | 26.16 | 9.10 | 9.73 | 25.96 | 43.18 | 15.09 | 9.25 | 14.51 | 15.32 | 11.43 | 10.73 | 25.84 | 8.76 | 37.86 |
| NUECTEN | 24.36 | 20.70 | 20.18 | 25.87 | 22.75 | 11.98 | 17.16 | 21.47 | 22.32 | 18.92 | 17.97 | 23.26 | 20.24 | 22.18 |
| YTFATEN | 68.87 | - | - | 61.25 | - | 12.65 | 70.47 | - | - | 68.92 | 66.13 | 50.25 | 66.84 | 70.83 |
| YTFSFLF | 15.83 | - | - | - | - | 14.85 | 9.80 | - | - | 10.11 | 10.92 | 16.32 | 9.27 | 16.05 |

*Table 2.* Multi-view Datasets Used in Experiments

| Dataset | SN | VN | Dimension | CN |
|---|---|---|---|---|
| FORETYP | 523 | 3 | 9/9/9 | 4 |
| CITSER | 3312 | 4 | 3312/3312/3312/3703 | 6 |
| NUSWINE | 4095 | 5 | 128/64/73/225/144 | 33 |
| NUECTEN | 6251 | 5 | 226/129/65/74/145 | 19 |
| YTFATEN | 38654 | 4 | 640/944/512/576 | 10 |
| YTFSFLF | 101499 | 5 | 647/64/64/838/512 | 31 |

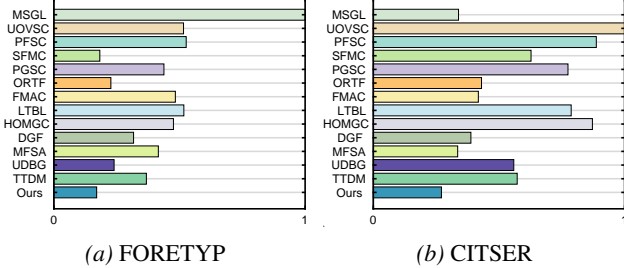

*(a)* FORETYP     *(b)* CITSER

*Figure 3.* Memory Usage Comparison

## 5. Experiments

To evaluate the clustering performance of FCFMVC, we conduct extensive experiments on six benchmark datasets. Their key characteristics are summarized in Table 2 where SN, VN, CN denote the number of samples, views and clusters, respectively. Then, the following thirteen algorithms are selected as baseline competitors for comparative analysis: MSGL (Kang et al., 2022), UOVSC (Tang et al., 2023), PFSC (Lv et al., 2021), SFMC (Li et al., 2022), PGSC (Wu et al., 2023), ORTF (Li et al., 2023a), FMAC (Wang et al., 2022), LTBL (Chen et al., 2023), HOMGC (Chen et al., 2024), DGF (Liang et al., 2024b), MFSA (Mei et al., 2025), UDBG (Fang et al., 2024), TTDM (Wang et al., 2025b). Their brief introduction is provided in Appendix Section K.

### 5.1. Clustering Results and Discussions

From the clustering results in Table 1 , we can draw that

(1) Our FCFMVC delivers favorable results across multiple datasets. For instance, on YTFATEN, it achieves the best results, and maintains a stable top-two ranking on FORETYP, NUSWINE, and YTFSFLF. These results collectively verify the effectiveness of FCFMVC in handling MVC problems.

(2) Several baselines, such as UOVSC, PFSC, SFMC, PGSC, LTBL, etc, fail to run on large-scale datasets YTFATEN and YTFSFLT. In contrast, our FCFMVC not only operates properly on all datasets but also delivers competitive results, which underscores its broader applicability and scalability.

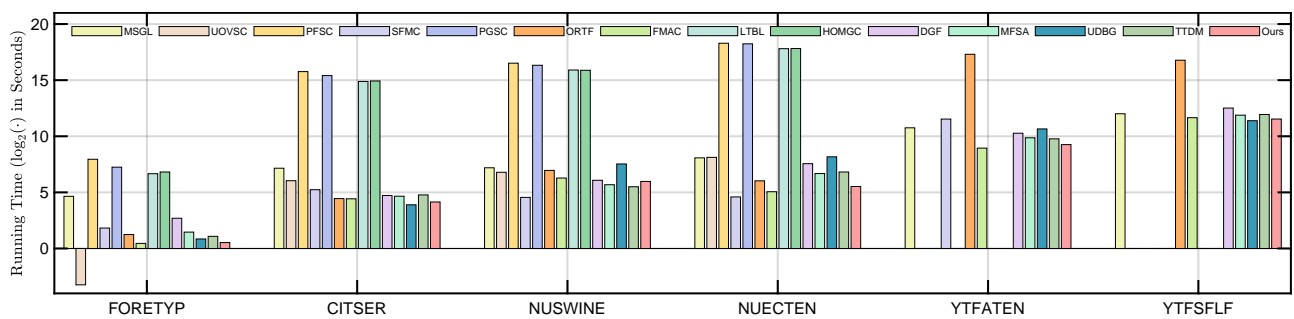

*Figure 4.* Time Usage Comparison

(3) FCFMVC yields somewhat inferior performance, such as on datasets CITSER and NUECTEN, possibly because the bipartite graph, while promoting view consistency, may over-smooth subtle yet discriminative sample–anchor relationships, accordingly diminishing structural distinctions.

## 5.2. Time and Space Usage Comparison

To demonstrate its resource-friendliness, we record the memory consumption and running time, as illustrated in Fig. 3 and 4. Evidently, our method incurs relatively low memory overhead and achieves favorable execution efficiency, which confirms that our method is efficient in tackling MVC tasks. More observations are provided in Appendix Section L.

*Table 3.* Ablation Results

| Metric | Ablation | DA-1 | DA-2 | DA-3 | DA-4 | DA-5 | DA-6 |
|--------|----------|------|------|------|------|------|------|
| ACC | WOB | 68.21 | 40.26 | 27.35 | 19.63 | 69.43 | 17.68 |
| | OSC | 75.32 | 45.53 | 35.63 | 22.74 | 72.87 | 21.63 |
| | OAC | 73.67 | 46.72 | 32.76 | 20.83 | 75.34 | 22.57 |
| | Ours | **82.79** | **49.58** | **38.62** | **25.13** | **78.29** | **26.32** |
| NMI | WOB | 48.73 | 16.52 | 7.67 | 8.54 | 71.36 | 15.84 |
| | OSC | 57.46 | 18.87 | 8.93 | 8.85 | 77.93 | 19.26 |
| | OAC | 54.37 | 19.56 | 8.19 | 8.97 | 75.31 | 17.86 |
| | Ours | **63.42** | **24.93** | **11.08** | **11.12** | **83.21** | **23.67** |
| PURITY | WOB | 62.57 | 30.63 | 28.47 | 25.83 | 61.63 | 25.74 |
| | OSC | 71.72 | 37.54 | 32.78 | 27.63 | 70.67 | 28.35 |
| | OAC | 74.52 | 39.28 | **39.71** | 27.26 | 74.38 | 27.82 |
| | Ours | **81.23** | **45.87** | 39.58 | **34.16** | **80.18** | **34.02** |
| FSCORE | WOB | 63.57 | 29.58 | 28.92 | 14.76 | 61.78 | 10.13 |
| | OSC | 66.48 | 32.74 | 30.16 | 16.83 | 64.82 | 12.74 |
| | OAC | 66.17 | 36.37 | 32.73 | 15.39 | 66.11 | 11.97 |
| | Ours | **72.17** | **40.24** | **37.86** | **22.18** | **70.83** | **16.05** |

## 5.3. Ablation Study

Table 3 presents the ablation study results, where OSC, OAC, WOB denote the configurations with only sample-cluster learning, only anchor-cluster learning, without both of them, respectively. DA-1~DA-6 refer to the datasets listed in Table 2. One can observe that both learning components collaboratively facilitate clustering performance, validating their complementary roles and joint contributions.

## 5.4. Convergence and Sensitivity

Fig. 5 and 6 depict the loss evolution and performance variation, respectively. As seen, the loss value declines consistently until reaching a steady level, while the performance curve fluctuates within a narrow range, which demonstrates that our devised method is convergent and relatively robust.

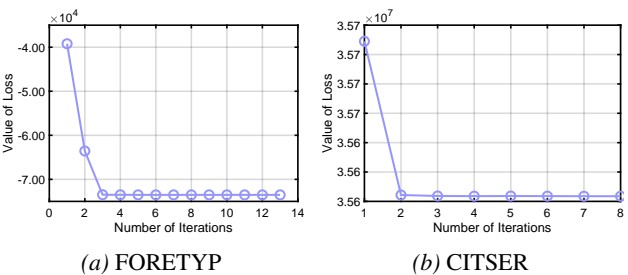

*(a)* FORETYP          *(b)* CITSER

*Figure 5.* Convergence Curve

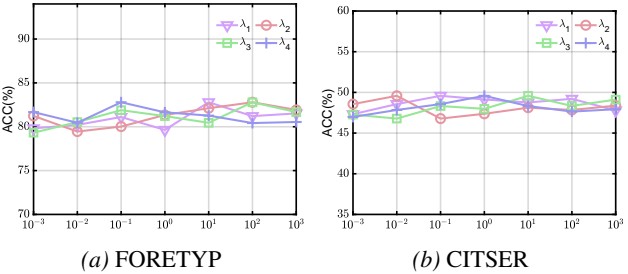

*(a)* FORETYP          *(b)* CITSER

*Figure 6.* Sensitivity Curve

## 6. Limitations and Concluding Remarks

In this work, we design a multi-view clustering approach termed FCFMVC to guide anchor allocation from anchor-level to cluster-level structures. Despite its effectiveness, FCFMVC involves hyper-parameters, which somewhat limits its practicality. Accordingly, developing a parameter-free variant will extend its applicability. Furthermore, it assumes full view completeness, failing to handle incomplete scenarios where partial view data is missing. This narrows its usability in real-world scenes with imperfect data collection.

## Impact Statement

This paper presents work whose goal is to advance the field of machine learning. There are many potential societal consequences of our work, none of which we feel must be specifically highlighted here.

## Acknowledgments

This work was supported in part by the Research Grants Council (RGC) Joint Research Scheme under grant: N_HKBU214/21, the General Research Fund of RGC under the grants: 12202622 and 12202924, the RGC Senior Research Fellow Scheme under grant: SRFS2324-2S02.

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

# Appendix

## A. Symbols

For greater clarity in the following complexity analyses and algorithmic derivations, we collate critical notations and their corresponding descriptions, as detailed in Table 4.

*Table 4.* List of Symbols and Their Descriptions

| Symbol | Size | Description |
|---|---|---|
| $n$ | Scalar | Sample size |
| $k$ | Scalar | Cluster size |
| $m$ | Scalar | Anchor size |
| $d_v$ | Scalar | Data dimension on view $v$ |
| $V$ | Scalar | View size |
| $\mathcal{C}_i$ | - | Cluster $i$ |
| $n_{\mathcal{C}_i}$ | Scalar | Number of samples in $\mathcal{C}_i$ |
| $\mathbf{c}_j$ | $m \times 1$ | Centroiod of cluster $j$ |
| $d_{i,\mathcal{C}_j}$ | Scalar | Similarity distance from sample $i$ to $\mathcal{C}_j$ |
| $\mu_{\mathcal{C}_j}$ | Scalar | Mean of similarity distance within cluster $j$ |
| $\sigma^2_{\mathcal{C}_j}$ | Scalar | Variance of similarity distance within cluster $j$ |
| $\frac{\bar{\sigma}}{\bar{\mu}}$ | Scalar | $\frac{1}{k}\sum_{j=1}^{k} \frac{\sigma_{\mathcal{C}_j}}{\mu_{\mathcal{C}_j}}$ |
| $h_j$ | Scalar | $\frac{1}{m}\sum_{i=1}^{m} [\mathbf{G}_v]_{ij}$ |
| $\frac{\sigma_{\mathcal{C}_r}}{\mu_{\mathcal{C}_r}}$ | Scalar | Cluster state before sample reassignment |
| $\frac{\sigma'_{\mathcal{C}_r}}{\mu'_{\mathcal{C}_r}}$ | Scalar | Cluster state after sample reassignment |
| $\mathbf{A}_v$ | $d_v \times m$ | Anchor matrix on view $v$ |
| $\mathbf{Z}$ | $m \times n$ | Bipartite graph |
| $\mathbf{F}$ | $n \times k$ | Sample-to-cluster indicator matrix |
| $\mathbf{G}_v$ | $m \times k$ | Anchor-to-cluster indicator matrix on view $v$ |
| $\mathbf{D}$ | $n \times m$ | $\sum_{v=1}^{V} \mathbf{X}_v^\top \mathbf{A}_v$ |
| $\mathbf{H}$ | $n \times m$ | $\mathbf{F}\sum_{v=1}^{V} \mathbf{G}_v^\top$ |
| $\mathbf{R}$ | $n \times k$ | $\mathbf{F} - \mathbf{Z}^\top \mathbf{G}_v$ |

## B. Proof of Theorem 1

*Proof.* We begin by deriving an explicit expression for the entries of the matrix product $\mathbf{F}^\top \mathbf{S} \mathbf{F}$. For any $p, q \in \{1, 2, \cdots, k\}$, the $(p, q)$ entry satisfies

$$\left(\mathbf{F}^\top \mathbf{S} \mathbf{F}\right)_{pq} = \sum_{a=1}^{n}\sum_{b=1}^{n} F_{pa}^\top S_{ab} F_{bq} = \sum_{a=1}^{n}\sum_{b=1}^{n} F_{ap} S_{ab} F_{bq}. \tag{28}$$

Recall that the trace of a square matrix is defined as the sum of its diagonal entries. Applying this definition to the matrix $\mathbf{F}^\top \mathbf{S} \mathbf{F}$ yields

$$\text{Tr}\left(\mathbf{F}^\top \mathbf{S} \mathbf{F}\right) = \sum_{i=1}^{k}\left(\mathbf{F}^\top \mathbf{S} \mathbf{F}\right)_{ii} = \sum_{i=1}^{k}\sum_{a=1}^{n}\sum_{b=1}^{n} F_{ai} S_{ab} F_{bi}. \tag{29}$$

Let $\mathcal{T}_i = \{a \subseteq \{1, 2, \cdots, n\} \mid F_{ai} = 1\}$ be the set of row indices corresponding to the nonzero (i.e., unit) entries in the $i$-th column of $\mathbf{F}$. Since $\mathbf{F}$ is a binary matrix satisfying $\mathbf{F}\mathbf{1}_k = \mathbf{1}_n$, each row of $\mathbf{F}$ contains exactly one non-zero entry.

Consequently, the product $F_{ai}F_{bi} \neq 0$ if and only if both $a$ and $b$ belong to $\mathcal{T}_i$. Therefore, we can derive

$$\text{Tr}\left(\mathbf{F}^\top \mathbf{S}\mathbf{F}\right) = \sum_{i=1}^{k} \sum_{a \in \mathcal{T}_i} \sum_{b \in \mathcal{T}_i} S_{ab}. \tag{30}$$

$\square$

## C. Proof of Theorem 2

*Proof.* By applying **Theorem 1**, we can obtain

$$\max_{\mathbf{Z}} \text{Tr}\left(\mathbf{F}^\top \mathbf{Z}^\top \mathbf{Z}\mathbf{F}\right) \Leftrightarrow \max_{\mathbf{Z}} \sum_{i=1}^{k} \sum_{a \in \mathcal{T}_i} \sum_{b \in \mathcal{T}_i} \left[\mathbf{Z}^\top \mathbf{Z}\right]_{ab}. \tag{31}$$

Using the definition of matrix multiplication, we have $[\mathbf{Z}^\top \mathbf{Z}]_{ab} = \mathbf{Z}_{:,a}^\top \mathbf{Z}_{:,b}$. Substituting this into the above expression yields

$$\max_{\mathbf{Z}} \text{Tr}\left(\mathbf{F}^\top \mathbf{Z}^\top \mathbf{Z}\mathbf{F}\right) \Leftrightarrow \max_{\mathbf{Z}} \sum_{i=1}^{k} \sum_{a \in \mathcal{T}_i} \sum_{b \in \mathcal{T}_i} \mathbf{Z}_{:,a}^\top \mathbf{Z}_{:,b}. \tag{32}$$

This expression shows that the original maximization is equivalent to maximizing, for each subset $\mathcal{T}_i$, the sum of all pairwise inner products among the columns of $\mathbf{Z}$ indexed by $\mathcal{T}_i$. That is, it equivalently requires maximizing the sum of $\mathbf{Z}_{:,a}^\top \mathbf{Z}_{:,b}$ on each $\mathcal{T}_i$. Consequently, the problem decouples across the disjoint column subsets,

$$\max_{\mathbf{Z}} \text{Tr}\left(\mathbf{F}^\top \mathbf{Z}^\top \mathbf{Z}\mathbf{F}\right) \Leftrightarrow \max_{\mathbf{Z}_{:,\mathcal{T}_i}} \sum_{a \in \mathcal{T}_i} \sum_{b \in \mathcal{T}_i} \mathbf{Z}_{:,a}^\top \mathbf{Z}_{:,b}, \quad i = 1, 2, \cdots, k. \tag{33}$$

We now analyze the double sum structure by partitioning the index pairs $(a, b) \in \mathcal{T}_i \times \mathcal{T}_i$. We can decompose $\sum_{a \in \mathcal{T}_i} \sum_{b \in \mathcal{T}_i} \mathbf{Z}_{:,a}^\top \mathbf{Z}_{:,b}$ into four components,

$$\sum_{a \in \mathcal{T}_i} \sum_{b \in \mathcal{T}_i} \mathbf{Z}_{:,a}^\top \mathbf{Z}_{:,b} = \sum_{a=j} \sum_{b=j} \mathbf{Z}_{:,a}^\top \mathbf{Z}_{:,b} + \sum_{a \in \mathcal{T}_i \setminus \{j\}} \sum_{b=j} \mathbf{Z}_{:,a}^\top \mathbf{Z}_{:,b} + \sum_{a=j} \sum_{b \in \mathcal{T}_i \setminus \{j\}} \mathbf{Z}_{:,a}^\top \mathbf{Z}_{:,b} + \sum_{a \in \mathcal{T}_i \setminus \{j\}} \sum_{b \in \mathcal{T}_i \setminus \{j\}} \mathbf{Z}_{:,a}^\top \mathbf{Z}_{:,b}. \tag{34}$$

By symmetry of the inner product and the summation indices, we have that the second and third terms are equal,

$$\sum_{a \in \mathcal{T}_i \setminus \{j\}} \sum_{b=j} \mathbf{Z}_{:,a}^\top \mathbf{Z}_{:,b} = \sum_{a=j} \sum_{b \in \mathcal{T}_i \setminus \{j\}} \mathbf{Z}_{:,a}^\top \mathbf{Z}_{:,b}. \tag{35}$$

Therefore, the optimization over $\mathbf{Z}_{:,\mathcal{T}_i}$ transforms to

$$\max_{\mathbf{Z}_{:,\mathcal{T}_i}} \sum_{a \in \mathcal{T}_i} \sum_{b \in \mathcal{T}_i} \mathbf{Z}_{:,a}^\top \mathbf{Z}_{:,b} \Leftrightarrow \max_{\mathbf{Z}_{:,j}} \sum_{a=j} \sum_{b=j} \mathbf{Z}_{:,a}^\top \mathbf{Z}_{:,b} + 2 \sum_{a \in \mathcal{T}_i \setminus \{j\}} \sum_{b=j} \mathbf{Z}_{:,a}^\top \mathbf{Z}_{:,b}, \quad \forall j \in \mathcal{T}_i \tag{36}$$

Since $\{\mathcal{T}_i\}_{i=1}^{k}$ are pairwise disjoint and satisfy $\sum_{i=1}^{k} |\mathcal{T}_i| = n$ where $|\mathcal{T}_i|$ denotes the size of set $\mathcal{T}_i$, we can equivalently transform the global maximization to independent per-column problems,

$$\max_{\mathbf{Z}} \text{Tr}\left(\mathbf{F}^\top \mathbf{Z}^\top \mathbf{Z}\mathbf{F}\right) \Leftrightarrow \max_{\mathbf{Z}_{:,j}} \mathbf{Z}_{:,j}^\top \mathbf{Z}_{:,j} + 2 \sum_{a \in \mathcal{T}_i \setminus \{j\}} \mathbf{Z}_{:,a}^\top \mathbf{Z}_{:,j} \Leftrightarrow \max_{\mathbf{Z}_{:,j}} \mathbf{Z}_{:,j}^\top \left(\mathbf{Z}_{:,j} + 2 \sum_{a \in \mathcal{T}_i \setminus \{j\}} \mathbf{Z}_{:,a}\right), \tag{37}$$

$$\forall j \in \mathcal{T}_i; \ \ i = 1, 2, \cdots, k,$$

which reveals that the original trace maximization decouples into independent (but block-coupled) quadratic sub-problems.

$\square$

## D. Proof of Theorem 3

*Proof.* Denote $\mathbf{E} = a\mathbf{I}_m + b\sum_{v=1}^{V} \mathbf{G}_v\mathbf{G}_v^\top$. Given $\mathbf{G}_v \in \{0,1\}^{m \times k}$ and $\mathbf{G}_v\mathbf{1}_k = \mathbf{1}_m$, we have $[\mathbf{G}_v\mathbf{G}_v^\top]_{ii} = 1$. Hence, the diagonal elements satisfy

$$E_{ii} = a + bV. \tag{38}$$

For $\mathbf{E}$ to be the zero matrix, all its diagonal and off-diagonal entries must vanish. Setting the diagonal entries to zero yields

$$a = -bV. \tag{39}$$

For non-diagonal elements $E_{ij}$ $(i \neq j)$, we have

$$E_{ij} = a \cdot [\mathbf{I}_m]_{ij} + b\sum_{v=1}^{V} \left[\mathbf{G}_v\mathbf{G}_v^\top\right]_{ij} = b\sum_{v=1}^{V} \left[\mathbf{G}_v\mathbf{G}_v^\top\right]_{ij}. \tag{40}$$

Thus, $E_{ij} = 0$ indicates

$$b = 0, \text{ or } \sum_{v=1}^{V} \left[\mathbf{G}_v\mathbf{G}_v^\top\right]_{ij} = 0. \tag{41}$$

- If $b = 0$, then according to Eq. (39) we have $a = 0$, which is conflict with the given conditions that $a$ and $b$ are not both zero.

- If $\sum_{v=1}^{V} \left[\mathbf{G}_v\mathbf{G}_v^\top\right]_{ij} = 0$, then for all $v$, $\left[\mathbf{G}_v\mathbf{G}_v^\top\right]_{ij} = 0$. This indicates that anchor $i$ and anchor $j$ are divided into different clusters for all $i \neq j$ in every view. However, since the number of clusters $k$ is less than the number of anchors $m$, we can obtain that in each view, at least one cluster contains at least two anchors. Consequently, there exist distinct indices $i \neq j$ such that $\left[\mathbf{G}_v\mathbf{G}_v^\top\right]_{ij} \neq 0$, contradicting the requirement that all off-diagonal sums vanish.

Therefore, under the stated conditions, the matrix $a\mathbf{I}_m + b\sum_{v=1}^{V} \mathbf{G}_v\mathbf{G}_v^\top$ is always non-zero.

$\square$

## E. Proof of Theorem 4

*Proof.* Using square expansion property of the Frobenius norm, we can obtain

$$\left\|\mathbf{F} - \mathbf{Z}^\top\mathbf{G}_v\right\|_F^2 = \left\|\mathbf{Z}^\top\mathbf{G}_v\right\|_F^2 - 2\langle\mathbf{F}, \mathbf{Z}^\top\mathbf{G}_v\rangle + n, \tag{42}$$

which suggests that the change in $\left\|\mathbf{F} - \mathbf{Z}^\top\mathbf{G}_v\right\|_F^2$ is primarily driven by $\mathbf{Z}^\top\mathbf{G}_v$, since both $\mathbf{Z}$ and $\mathbf{F}$ are held fixed during the optimization of $\mathbf{G}_v$.

Denote $\mathbf{p}_i = \mathbf{Z}_{i,:}^\top$, i.e., the transpose of the $i$-th row of $\mathbf{Z}$. Then, we have $\mathbf{Z}^\top = [\mathbf{p}_1, \mathbf{p}_2, \cdots, \mathbf{p}_m]$. Combined with $\mathbf{G}_v = \left[[\mathbf{G}_v]_{1,:}; [\mathbf{G}_v]_{2,:}; \cdots; [\mathbf{G}_v]_{m,:}\right]$, we can get

$$\mathbf{Z}^\top\mathbf{G}_v = \sum_{i=1}^{m} \mathbf{p}_i \cdot [\mathbf{G}_v]_{i,:}. \tag{43}$$

When reassigning anchor $j$ from cluster $c$ to cluster $l$, only two entries of $\mathbf{G}_v$ changes. Specially, $[\mathbf{G}_v]_{j,c}$ transitions from 1 to 0, $[\mathbf{G}_v]_{j,l}$ transitions from 0 to 1, while all other entries maintain unchanged. Therefore, we have that the change in $\mathbf{Z}^\top\mathbf{G}_v$ is

$$\mathbf{\Delta} = \mathbf{p}_j \cdot (\mathbf{e}_l - \mathbf{e}_c)^\top, \tag{44}$$

where $\mathbf{e}_l$ and $\mathbf{e}_c$ are $k$-dimensional standard basis vectors.

On the basis of this, we can derive

$$\left\|\mathbf{F} - \left(\mathbf{Z}^\top\mathbf{G}_v + \mathbf{\Delta}\right)\right\|_F^2 - \left\|\mathbf{F} - \mathbf{Z}^\top\mathbf{G}_v\right\|_F^2 = \|\mathbf{\Delta}\|_F^2 - 2\langle\mathbf{F} - \mathbf{Z}^\top\mathbf{G}_v, \mathbf{\Delta}\rangle. \tag{45}$$

Combined with the fact that $\|\mathbf{AB}\|_F^2 \le \|\mathbf{A}\|_F^2 \cdot \|\mathbf{B}\|_F^2$ where the equality holds if and only if $\mathbf{A}$ or $\mathbf{B}$ is a rank-1 matrix, we can obtain

$$\|\mathbf{\Delta}\|_F^2 = \|\mathbf{p}_j\|^2 \cdot \|\mathbf{e}_l - \mathbf{e}_c\|^2 = 2\|\mathbf{p}_j\|^2. \tag{46}$$

Besides, for the inner-product term $\langle \mathbf{F} - \mathbf{Z}^\top \mathbf{G}_v, \mathbf{\Delta} \rangle$, via element-wise expansion, we have

$$\langle \mathbf{F} - \mathbf{Z}^\top \mathbf{G}_v, \mathbf{\Delta} \rangle = \sum_{q=1}^{n} \sum_{g=1}^{k} \left( F_{qg} - \left[ \mathbf{Z}^\top \mathbf{G}_v \right]_{qg} \right) \cdot \left( p_j^q \cdot (\mathbf{e}_l - \mathbf{e}_c)_g \right), \tag{47}$$

where $p_j^q$ and $(\mathbf{e}_l - \mathbf{e}_c)_g$ denote the $q$-th and $g$-th component of $\mathbf{p}_j$ and $(\mathbf{e}_l - \mathbf{e}_c)$, respectively.

Since

$$(\mathbf{e}_l - \mathbf{e}_c)_g = \begin{cases} 1, & g = l \\ -1, & g = c \\ 0, & \text{otherwise,} \end{cases} \tag{48}$$

we can get

$$\langle \mathbf{F} - \mathbf{Z}^\top \mathbf{G}_v, \mathbf{\Delta} \rangle = \sum_{q=1}^{n} \left[ \left( F_{ql} - \left[ \mathbf{Z}^\top \mathbf{G}_v \right]_{ql} \right) p_j^q - \left( F_{qc} - \left[ \mathbf{Z}^\top \mathbf{G}_v \right]_{qc} \right) p_j^q \right]. \tag{49}$$

Therefore, we have

$$\langle \mathbf{F} - \mathbf{Z}^\top \mathbf{G}_v, \mathbf{\Delta} \rangle = \mathbf{p}_j^\top \left( \left[ \mathbf{F} - \mathbf{Z}^\top \mathbf{G}_v \right]_{:,l} - \left[ \mathbf{F} - \mathbf{Z}^\top \mathbf{G}_v \right]_{:,c} \right). \tag{50}$$

Combining Eqs. (45), (46) and (50), and recalling that $\mathbf{p}_j = \mathbf{Z}_{j,:}^\top$, we can obtain the change in $\left\| \mathbf{F} - \mathbf{Z}^\top \mathbf{G}_v \right\|_F^2$ is

$$\begin{aligned} \Delta_{zgf} &= 2\|\mathbf{Z}_{j,:}\|^2 - 2\mathbf{Z}_{j,:} \left( \left[ \mathbf{F} - \mathbf{Z}^\top \mathbf{G}_v \right]_{:,l} - \left[ \mathbf{F} - \mathbf{Z}^\top \mathbf{G}_v \right]_{:,c} \right) \\ &= 2\mathbf{Z}_{j,:} \left( \mathbf{Z}_{j,:}^\top + \left[ \mathbf{F} - \mathbf{Z}^\top \mathbf{G}_v \right]_{:,c} - \left[ \mathbf{F} - \mathbf{Z}^\top \mathbf{G}_v \right]_{:,l} \right). \end{aligned} \tag{51}$$

$\square$

## F. Proof of Theorem 5

*Proof.* To establish **Theorem 5**, we first let

$$b_j = \frac{\sigma_{\mathcal{C}_j}}{\mu_{\mathcal{C}_j}}, \quad \bar{b} = \frac{\bar{\sigma}}{\bar{\mu}}, \tag{52}$$

and then we have

$$\frac{1}{k} \sum_{j=1}^{k} \left( \frac{\sigma_{\mathcal{C}_j}}{\mu_{\mathcal{C}_j}} - \frac{\bar{\sigma}}{\bar{\mu}} \right)^2 = \frac{1}{k} \sum_{j=1}^{k} \left( b_j - \bar{b} \right)^2 = \frac{1}{k} \sum_{j=1}^{k} b_j^2 - \bar{b}^2. \tag{53}$$

For each cluster $j$, we define two cluster-wise summary statistics,

$$D1_j = \sum_{i \in \mathcal{C}_j} d_{i,\mathcal{C}_j}, \quad D2_j = \sum_{i \in \mathcal{C}_j} d_{i,\mathcal{C}_j}^2 \tag{54}$$

In conjunction with the meaning of $\mu_{\mathcal{C}_j}$ and $\sigma_{\mathcal{C}_j}^2$, we can obtain

$$\mu_{\mathcal{C}_j} = \frac{D1_j}{n_{\mathcal{C}_j}}, \quad \sigma_{\mathcal{C}_j}^2 = \frac{D2_j}{n_{\mathcal{C}_j}} - \mu_{\mathcal{C}_j}^2. \tag{55}$$

Based on this, we can further obtain

$$b_j = \frac{\sqrt{\sigma_{\mathcal{C}_j}^2}}{\mu_{\mathcal{C}_j}} = \frac{\sqrt{\frac{D2_j}{n_{\mathcal{C}_j}} - \left( \frac{D1_j}{n_{\mathcal{C}_j}} \right)^2}}{\frac{D1_j}{n_{\mathcal{C}_j}}} = \frac{\sqrt{n_{\mathcal{C}_j} D2_j - D1_j^2}}{D1_j}. \tag{56}$$

Kindly note that in practice $D1_j$ is greater than 0. $D1_j = 0$, i.e., $\|\mathbf{z}_i - \mathbf{c}_j\| = 0$ for any $i \in \mathcal{C}_j$, indicates that all samples in cluster $j$ are located at the centroid, which is almost never occurs in a continuous space.

When reassigning sample $j$ from cluster $r$ to cluster $t$, it needs to update $\mathbf{c}_r$, $d_{j,\mathcal{C}_r}$, $\mathbf{c}_t$ and $d_{j,\mathcal{C}_t}$ since the centroids are changed. Accordingly, we have

$$b'_r \neq b_r, \; b'_t \neq b_t, \; b'_p = b_p \; (\forall p \notin \{r,t\}) \tag{57}$$

and

$$\bar{b}' = \frac{1}{k} \left[ b'_r + b'_t + \sum_{p \notin \{r,t\}} b_p \right] = \bar{b} + \frac{1}{k} \left[ (b'_r - b_r) + (b'_t - b_t) \right], \tag{58}$$

where $b_r$ denotes the state before moving the sample while $b'_r$ denotes the state after moving the sample.

Define

$$\widehat{\Delta}_r = b'_r - b_r, \quad \widehat{\Delta}_t = b'_t - b_t, \quad \xi = \frac{\widehat{\Delta}_r + \widehat{\Delta}_t}{k}, \tag{59}$$

and then, we have

$$\bar{b}' = \bar{b} + \xi. \tag{60}$$

Denote

$$R = \frac{1}{k} \sum_{j=1}^{k} \left( \frac{\sigma_{\mathcal{C}_j}}{\mu_{\mathcal{C}_j}} - \frac{\bar{\sigma}}{\bar{\mu}} \right)^2 = \frac{1}{k} \sum_{j=1}^{k} \left( \bar{b} - b_j \right)^2, \tag{61}$$

and after moving the sample, we have

$$R' = \frac{1}{k} \left[ (\bar{b}' - b'_r)^2 + (\bar{b}' - b'_t)^2 + \sum_{p \notin \{r,t\}} (\bar{b}' - b_p)^2 \right]. \tag{62}$$

Note that for any $p \notin \{r,t\}$, we have

$$\bar{b}' - b_p = \bar{b} - b_p + \xi. \tag{63}$$

Therefore, combined with Eqs. (59), (60) and (63), we can get

$$R' = \frac{1}{k} \left[ \left( \bar{b} + \xi - b_r - \widehat{\Delta}_r \right)^2 + \left( \bar{b} + \xi - b_t - \widehat{\Delta}_t \right)^2 + \sum_{p \notin \{r,t\}} (\bar{b} - b_p + \xi)^2 \right]. \tag{64}$$

Denote current deviation $d_j = \bar{b} - b_j$. Then, we have

$$\left( \bar{b} + \xi - b_r - \widehat{\Delta}_r \right)^2 = \left( d_r + \xi - \widehat{\Delta}_r \right)^2,$$
$$\left( \bar{b} + \xi - b_t - \widehat{\Delta}_t \right)^2 = \left( d_t + \xi - \widehat{\Delta}_t \right)^2, \tag{65}$$
$$(\bar{b} - b_p + \xi)^2 = (d_p + \xi)^2.$$

Combining Eqs. (61), (64) and (65) yields

$$R' - R = \frac{1}{k} \left[ \left( d_r + \xi - \widehat{\Delta}_r \right)^2 - d_r^2 + \left( d_t + \xi - \widehat{\Delta}_t \right)^2 - d_t^2 + \sum_{p \notin \{r,t\}} \left( (d_p + \xi)^2 - d_p^2 \right) \right]. \tag{66}$$

For the difference terms, we have

$$\left( d_r + \xi - \widehat{\Delta}_r \right)^2 - d_r^2 = (\xi - \widehat{\Delta}_r)(2d_r + \xi - \widehat{\Delta}_r),$$
$$\left( d_t + \xi - \widehat{\Delta}_t \right)^2 - d_t^2 = (\xi - \widehat{\Delta}_t)(2d_t + \xi - \widehat{\Delta}_t), \tag{67}$$
$$(d_p + \xi)^2 - d_p^2 = \xi \, (2d_p + \xi).$$

Note that

$$\sum_{p \notin \{r,t\}} d_p = \sum_{p=1}^{k} d_p - d_r - d_t, \quad \sum_{p=1}^{k} d_p = \sum_{p=1}^{k} (\bar{b} - b_p) = k\bar{b} - \sum_{p=1}^{k} b_p = 0, \tag{68}$$

and therefore we can get

$$\sum_{p \notin \{r,t\}} d_p = -d_r - d_t. \tag{69}$$

Accordingly, we have

$$\sum_{p \notin \{r,t\}} \xi \left( 2d_p + \xi \right) = \xi \left[ 2(-d_r - d_t) + (k-2)\xi \right]. \tag{70}$$

Combined with Eqs. (66), (67) and (70), we can get

$$R' - R = \frac{1}{k} \left[ (\xi - \widehat{\Delta}_r)(2d_r + \xi - \widehat{\Delta}_r) + (\xi - \widehat{\Delta}_t)(2d_t + \xi - \widehat{\Delta}_t) + \xi \left[ (k-2)\xi - 2(d_r + d_t) \right] \right]. \tag{71}$$

Denote

$$\begin{aligned}
A_r &= (\xi - \widehat{\Delta}_r)(2d_r + \xi - \widehat{\Delta}_r) = 2d_r\xi - 2d_r\widehat{\Delta}_r + \xi^2 - 2\xi\widehat{\Delta}_r + \widehat{\Delta}_r^2, \\
A_t &= (\xi - \widehat{\Delta}_t)(2d_t + \xi - \widehat{\Delta}_t) = 2d_t\xi - 2d_t\widehat{\Delta}_t + \xi^2 - 2\xi\widehat{\Delta}_t + \widehat{\Delta}_t^2, \\
B &= \xi[(k-2)\xi - 2(d_r + d_t)] = (k-2)\xi^2 - 2d_r\xi - 2d_t\xi,
\end{aligned} \tag{72}$$

and then we have

$$A_r + A_t + B = \widehat{\Delta}_r^2 + \widehat{\Delta}_t^2 - 2d_r\widehat{\Delta}_r - 2d_t\widehat{\Delta}_t + k\xi^2 - 2\xi \left( \widehat{\Delta}_r + \widehat{\Delta}_t \right). \tag{73}$$

In conjunction with $\xi = \frac{\widehat{\Delta}_r + \widehat{\Delta}_t}{k}$, we have

$$k\xi^2 = \frac{\left( \widehat{\Delta}_r + \widehat{\Delta}_t \right)^2}{k}. \tag{74}$$

Therefore, we can get

$$k\xi^2 - 2\xi \left( \widehat{\Delta}_r + \widehat{\Delta}_t \right) = -\frac{\left( \widehat{\Delta}_r + \widehat{\Delta}_t \right)^2}{k}. \tag{75}$$

Combined with Eqs. (71), (72), (73) and (75), we can get

$$\begin{aligned}
\Delta_{cv} = R' - R &= \frac{1}{k} \left[ \widehat{\Delta}_r^2 + \widehat{\Delta}_t^2 - \frac{\left( \widehat{\Delta}_r + \widehat{\Delta}_t \right)^2}{k} - 2 \left( d_r\widehat{\Delta}_r + d_t\widehat{\Delta}_t \right) \right] \\
&= \frac{1}{k^2} \left( (k-1) \left( \widehat{\Delta}_r^2 + \widehat{\Delta}_t^2 \right) - 2\widehat{\Delta}_r\widehat{\Delta}_t - 2k \left( d_r\widehat{\Delta}_r + d_t\widehat{\Delta}_t \right) \right).
\end{aligned} \tag{76}$$

(where $d_r = \bar{b} - b_r$, $d_t = \bar{b} - b_t$, $\bar{b} = \frac{1}{k}\sum_{j=1}^{k} b_j$, $b_j = \frac{\sigma_{\mathcal{C}_j}}{\mu_{\mathcal{C}_j}}$, $\widehat{\Delta}_r = b'_r - b_r$, $\widehat{\Delta}_t = b'_t - b_t$. It requires calculating $b'_r$ and $b'_t$ since $b_j$ has already been given for $j = 1, 2, \cdots, k$. )

The centroids do undergo a change upon sample movement, and accordingly the $d_{i,\mathcal{C}_j}$ values corresponding to all samples in this cluster will change. Each time a sample is moved, the cost associated with precise updating is relatively high. To this end, we first treat the centroids as fixed entities in the current scan, and then update all centroids synchronously prior to the next round of scanning. That is, after a complete round of sample scanning, we recalculate all centroids for all clusters with altered member compositions in preparation for the next scanning round. Consequently, after movement, we have

$$\begin{aligned}
n'_{\mathcal{C}_r} &= n_{\mathcal{C}_r} - 1, \quad n'_{\mathcal{C}_t} = n_{\mathcal{C}_t} + 1, \\
D1'_r &= D1_r - d_{j,\mathcal{C}_r}, \quad D2'_r = D2_r - d_{j,\mathcal{C}_r}^2, \\
D1'_t &= D1_t + d_{j,\mathcal{C}_t}, \quad D2'_t = D2_t + d_{j,\mathcal{C}_t}^2.
\end{aligned} \tag{77}$$

For any $p \notin \{r, t\}$, note that the sample composition of cluster $p$ remains unaltered. Consequently, the aggregated statistics $D1_p$, $D2_p$, and the cluster size $n_{\mathcal{C}_p}$ retain their original values, and accordingly can be directly reused in subsequent calculations.

In conjunction with Eq. (56), we can get

$$b'_r = \frac{\sqrt{n'_{\mathcal{C}_r} D2'_r - (D1'_r)^2}}{D1'_r}, \quad b'_t = \frac{\sqrt{n'_{\mathcal{C}_t} D2'_t - (D1'_t)^2}}{D1'_t}. \tag{78}$$

On the basis of this, combining Eqs. (52), (54), (77), we can obtain the standard-deviation-to-mean ratio following the sample reassignment, as shown in Eq. (26).

Then, combining Eqs. (76) and (78), we can derive the specific change $\Delta_{cv}$ in $\frac{1}{k} \sum_{j=1}^{k} \left( \frac{\sigma_{\mathcal{C}_j}}{\mu_{\mathcal{C}_j}} - \frac{\bar{\sigma}}{\bar{\mu}} \right)^2$.

$\square$

# G. Proof of Theorem 6

*Proof.* **Computational Complexity:** The computational overhead mainly stems from the updates of $\mathbf{A}_v$, $\mathbf{Z}$, $\mathbf{G}_v$ and $\mathbf{F}$. During the update of $\mathbf{A}_v$, building the term $\mathbf{X}_v \mathbf{Z}^\top$ needs $\mathcal{O}(d_v nm)$ cost. The SVD of $\mathbf{X}_v \mathbf{Z}^\top$ needs $\mathcal{O}(d_v m^2)$. Combined with the fact $n$ is usually largely greater than $m$, therefore, the update of $\mathbf{A}_v$ needs $\mathcal{O}(d_v nm)$ cost.

During the update of $\mathbf{Z}$, building $\mathbf{D}$ and $\mathbf{H}$ takes $\mathcal{O}(ndm)$ and $\mathcal{O}(mkV + nkm)$ respectively, where $d = \sum_{v=1}^{V} d_v$. Building $\sum_{v=1}^{V} \mathbf{G}_v \mathbf{G}_v^\top$ takes $\mathcal{O}(m^2 kV)$. Each column of $\mathbf{Z}$ is successfully transformed into a quadratic programming problem, and accordingly can be solved within $\mathcal{O}(m^3)$. Instead of directly building $\sum_{a \in \mathcal{C}_i \setminus \{j\}} \mathbf{Z}_{:,a}^\top$ for each $\mathbf{Z}_{:,j}$, we can build $\sum_{a \in \mathcal{C}_i} \mathbf{Z}_{:,a}^\top$ for all $\mathbf{Z}_{:,j}$ in the same cluster, and then subtract current $\mathbf{Z}_{:,j}$ from it. Accordingly, the cost is $\mathcal{O}(mn_{\mathcal{C}_i})$. Building $\sum_{i=1}^{k} \sum_{a \in \mathcal{C}_i} \mathbf{Z}_{:,a}^\top$ needs $\sum_{i=1}^{k} \mathcal{O}(mn_{\mathcal{C}_i})$, i.e. $\mathcal{O}(mn)$. (When optimizing other $j$ in the same cluster, we can construct the term $\sum_{a \in \mathcal{C}_i \setminus \{j\}} \mathbf{Z}_{:,a}^\top$ by first utilizing $\left( \sum_{a \in \mathcal{C}_i} \mathbf{Z}_{:,a}^\top \right) - \mathbf{Z}_{:,j_{previous}}^\top + \mathbf{Z}_{:,j_{previous}}^{\top updated}$ to update $\left( \sum_{a \in \mathcal{C}_i} \mathbf{Z}_{:,a}^\top \right)$ and then performing $\left( \sum_{a \in \mathcal{C}_i} \mathbf{Z}_{:,a}^\top \right) - \mathbf{Z}_{:,j_{current}}^\top$ where $j_{previous}$ denotes the previous $j$ in the same cluster. For example, suppose cluster $i = 2$ consists of samples $\{1, 3, 5, 7\}$. Denote $S_{um} = \sum_{a \in \mathcal{C}_2} \mathbf{Z}_{:,a}^\top = \mathbf{Z}_{:,1}^\top + \mathbf{Z}_{:,3}^\top + \mathbf{Z}_{:,5}^\top + \mathbf{Z}_{:,7}^\top$. When optimizing $\mathbf{Z}_{:,1}$, we utilize $S_{um} - \mathbf{Z}_{:,1}^\top$ to construct $\sum_{a \in \mathcal{C}_i \setminus \{j\}} \mathbf{Z}_{:,a}^\top$. Then, with updated $\mathbf{Z}_{:,1^{updated}}$, we update $S_{um} = \left( S_{um} - \mathbf{Z}_{:,1}^\top \right) + \mathbf{Z}_{:,1^{updated}}^\top$. During optimizing $\mathbf{Z}_{:,3}$, we subtract $\mathbf{Z}_{:,3}^\top$ from the updated $S_{um}$ to construct $\sum_{a \in \mathcal{C}_i \setminus \{j\}} \mathbf{Z}_{:,a}^\top$. With updated $\mathbf{Z}_{:,3^{updated}}$, we update $S_{um} = \left( S_{um} - \mathbf{Z}_{:,3}^\top \right) + \mathbf{Z}_{:,3^{updated}}^\top$ to serve for the optimization of $\mathbf{Z}_{:,5}$. ) Consequently, we have that it totally takes $\mathcal{O}(ndm + nkm + m^2 kV + m^3 n + mn)$ to update $\mathbf{Z}$. In practice, the cluster size $k$ is generally smaller than the data dimension $d$, and the sample size $n$ is largely greater than the view size $V$. Therefore, the cost of updating $\mathbf{Z}$ can simplify to $\mathcal{O}(ndm + m^3 n)$.

During the update of $\mathbf{G}_v$, building $\mathbf{R}$ needs $\mathcal{O}(nmk)$ cost. Computing the value of function $f$ needs $\mathcal{O}(m)$ cost. Updating $\mathbf{R}$ and the cluster size proportions $h_c$ and $h_{l*}$ needs $\mathcal{O}(m)$ and constant costs, respectively. Correspondingly, scanning one row needs $\mathcal{O}(nmk)$ cost. Since $\mathbf{R}$ is reusable and only needs to update certain columns, we have that the update of $\mathbf{G}_v$ needs at most $\mathcal{O}(nm^2 k)$ computational cost.

During the update of $\mathbf{F}$, when scanning each row, rather than building $\sum_{a \in \mathcal{C}_r, a \neq j} \mathbf{Z}_{:,a}$ directly, we first build $\sum_{a \in \mathcal{C}_r} \mathbf{Z}_{:,a}$, and then subtract $\mathbf{Z}_{:,j}$ from it, which takes $\mathcal{O}(mn_{\mathcal{C}_r})$ cost. Building all $\sum_{a \in \mathcal{C}_r} \mathbf{Z}_{:,a}$ with $r = 1 \sim k$ needs $\mathcal{O}(mn)$. (Note that $\sum_{a \in \mathcal{C}_r} \mathbf{Z}_{:,a}$ with $r = 1 \sim k$ are applicable during the process of scanning all rows.) Building $\Delta_{fz}$ needs $\mathcal{O}(m + mn_{\mathcal{C}_t} + mn_{\mathcal{C}_r})$. Therefore, after scanning a row, building $\Delta_{fz}$ totally takes $\mathcal{O}(mk + mn)$. Then, building $\mathbf{Z}^\top \sum_{v=1}^{V} \mathbf{G}_v$ needs $\mathcal{O}(mkV + mnk)$, and accordingly building $\Delta_{fzg}$ needs $\mathcal{O}(mkV + mnk)$. Due to only involving scalar, building $\Delta_{si}$ needs constant cost. During the building of $\Delta_{cv}$, we first build $\sum_{i \in \mathcal{C}_r} d_{i,\mathcal{C}_r}$ and $\sum_{i \in \mathcal{C}_r} d_{i,\mathcal{C}_r}^2$, and then construct $\frac{\sigma'_{\mathcal{C}_r}}{\mu'_{\mathcal{C}_r}}$ and $\frac{\sigma'_{\mathcal{C}_t}}{\mu'_{\mathcal{C}_t}}$, which takes $\mathcal{O}(mn_{\mathcal{C}_r})$ and $\mathcal{O}(mn_{\mathcal{C}_t})$, respectively. Therefore, building all $\frac{\sigma'_{\mathcal{C}_r}}{\mu'_{\mathcal{C}_r}}$ with $r = 1 \sim k$ needs $\mathcal{O}(mn)$ cost. All $\frac{\sigma_{\mathcal{C}_r}}{\mu_{\mathcal{C}_r}}$ are applicable during optimization. Kindly note that during scanning each row, the established $\sum_{a \in \mathcal{C}_t} \mathbf{Z}_{:,a}$, $\mathbf{Z}^\top \sum_{v=1}^{V} \mathbf{G}_v$, $n_{\mathcal{C}_t}$, $\sum_{i \in \mathcal{C}_t} d_{i,\mathcal{C}_t}$ and $\sum_{i \in \mathcal{C}_t} d_{i,\mathcal{C}_t}^2$ with $t = 1 \sim k$ are reusable, and only needs simple addition or subtraction operation to adapt to the new state. During scanning each row, therefore, it totally needs $\mathcal{O}(mk + mn + mkV + mnk)$ cost. Based on this, we have that updating $\mathbf{F}$ takes $\mathcal{O}(mkn + mn + mkV + mnk)$ cost. Usually, the view size $V$ is largely

smaller than the sample size $n$. Therefore, the update of $\mathbf{F}$ needs $\mathcal{O}(mkn)$ cost.

In conjunction with the above analysis, we can obtain that the overall computational cost is $\mathcal{O}(dnm + m^3n + nm^2k)$. Due to the fact that the anchor size $m$ and the cluster size $k$ are largely smaller than the sample size $n$ and that the data dimension $d$ is a constant and irrelevant to $n$, we have that the computational complexity of our proposed algorithm is $\mathcal{O}(n)$, i.e., linear to the sample size.

**Space Complexity:** The space overhead mainly stems from the storage of variables $\mathbf{A}_v$, $\mathbf{Z}$, $\mathbf{G}_v$ and $\mathbf{F}$. During the update of $\mathbf{A}_v$, building the product $\mathbf{X}_v\mathbf{Z}^\top$ needs $\mathcal{O}(d_vm)$ space cost, and generating its SVD needs $\mathcal{O}(d_vm + m^2)$. Combined with the size of $\mathbf{A}_v \in \mathbb{R}^{d_v \times m}$, therefore, the update of $\mathbf{A}_v$ needs $\mathcal{O}(d_vm + m^2)$ space cost.

During the update of $\mathbf{Z}$, building $\mathbf{D}$, $\mathbf{H}$ and $\sum_{v=1}^{V} \mathbf{G}_v\mathbf{G}_v$ needs $\mathcal{O}(nmV)$, $\mathcal{O}(mkV + nm)$ and $\mathcal{O}(mV)$ space costs, respectively. Building $\sum_{a \in \mathcal{C}_i} \mathbf{Z}_{:,a}^\top$ with $i = 1 \sim k$ needs $\mathcal{O}(mk)$. Therefore, the space cost about updating $\mathbf{Z}$ is $\mathcal{O}(nmV + mkV)$.

During the update of $\mathbf{G}_v$, building $\mathbf{R}$ needs $\mathcal{O}(nk)$. Since the value of function $f$ is a scalar, we have that updating $\mathbf{G}_v$ needs $\mathcal{O}(nk)$ space cost.

During the update of $\mathbf{F}$, building $\sum_{a \in \mathcal{C}_r} \mathbf{Z}_{:,a}$ with $r = 1 \sim k$ and $\mathbf{Z}^\top \sum_{v=1}^{V} \mathbf{G}_v$ needs $\mathcal{O}(mk)$ and $\mathcal{O}(mkV + nk)$, respectively. Building $\sum_{i \in \mathcal{C}_r} d_{i,\mathcal{C}_r}$ and $\sum_{i \in \mathcal{C}_r} d_{i,\mathcal{C}_r}^2$ needs $\mathcal{O}(mk)$ and $\mathcal{O}(mk)$, respectively. Therefore, updating $\mathbf{F}$ needs $\mathcal{O}(mkV + nk)$ space cost.

Based on the above analysis, we have that the overall space cost is $\mathcal{O}(dm + m^2V + nmV + mkV + nk)$. Since $d$ is not relevant to $n$, and $m$ and $k$ are largely smaller than $n$, we can obtain that the space complexity of the proposed algorithm is $\mathcal{O}(n)$.

$\square$

## H. Derivation Steps

In this section, we provide more detailed derivation steps for addressing the problem (7). Its explicit expression is given as follows:

$$
\begin{aligned}
\min_{\mathbf{A}_v, \mathbf{Z}, \mathbf{F}, \mathbf{G}_v} & \left\{ \sum_{v=1}^{V} \|\mathbf{X}_v - \mathbf{A}_v\mathbf{Z}\|_F^2 - \lambda_1 \operatorname{Tr}\left(\mathbf{F}^\top \mathbf{Z}^\top \mathbf{Z} \mathbf{F}\right) + \lambda_2 \left( \left\| \frac{\mathbf{F}^\top \mathbf{F}}{n} - \frac{\mathbf{I}_k}{k} \right\|_F^2 + \frac{1}{k} \sum_{j=1}^{k} \left( \frac{\sigma_{\mathcal{C}_j}}{\mu_{\mathcal{C}_j}} - \frac{\bar{\sigma}}{\bar{\mu}} \right)^2 \right) + \right. \\
& \left. \lambda_3 \sum_{v=1}^{V} \|\mathbf{F} - \mathbf{Z}^\top \mathbf{G}_v\|_F^2 + \lambda_4 \sum_{v=1}^{V} \sum_{j=1}^{k} \left[ \max\left(0, \frac{\mathbf{1}_m^\top [\mathbf{G}_v]_{.,j}}{m} - \frac{1+\delta}{k}\right)^2 + \max\left(0, \frac{1-\delta}{k} - \frac{\mathbf{1}_m^\top [\mathbf{G}_v]_{.,j}}{m}\right)^2 \right] \right\} \\
\text{s.t. } & \mathbf{A}_v^\top \mathbf{A}_v = \mathbf{I}_m, \mathbf{Z} \geq 0, \mathbf{Z}^\top \mathbf{1}_m = \mathbf{1}_n, \mathbf{F} \in \{0,1\}^{n \times k}, \mathbf{F}\mathbf{1}_k = \mathbf{1}_n, \mathbf{G}_v \in \{0,1\}^{m \times k}, \mathbf{G}_v\mathbf{1}_k = \mathbf{1}_m.
\end{aligned}
\tag{79}
$$

### H.1. $\mathbf{A}_v$ Sub-problem

The problem concerning $\mathbf{A}_v$ is

$$
\begin{aligned}
\min_{\mathbf{A}_v} & \sum_{v=1}^{V} \|\mathbf{X}_v - \mathbf{A}_v\mathbf{Z}\|_F^2 \\
\text{s.t. } & \mathbf{A}_v^\top \mathbf{A}_v = \mathbf{I}_m.
\end{aligned}
\tag{80}
$$

In practice, view data are independent of each other, and accordingly anchors on different views are also mutually independent of each other. Therefore, Eq. (80) equivalently becomes

$$
\begin{aligned}
\min_{\mathbf{A}_v} & \|\mathbf{X}_v - \mathbf{A}_v\mathbf{Z}\|_F^2 \\
\text{s.t. } & \mathbf{A}_v^\top \mathbf{A}_v = \mathbf{I}_m.
\end{aligned}
\tag{81}
$$

Combined with $\|\mathbf{X}_v - \mathbf{A}_v\mathbf{Z}\|_F^2 = \operatorname{Tr}\left(\mathbf{X}_v\mathbf{X}_v^\top - 2\mathbf{X}_v\mathbf{Z}^\top\mathbf{A}_v^\top + \mathbf{Z}^\top\mathbf{Z}\right)$, we have

$$
\min_{\mathbf{A}_v} \|\mathbf{X}_v - \mathbf{A}_v\mathbf{Z}\|_F^2 \Leftrightarrow \max_{\mathbf{A}_v} \operatorname{Tr}\left(\mathbf{X}_v\mathbf{Z}^\top\mathbf{A}_v^\top\right).
\tag{82}
$$

Denote $\mathbf{U}\boldsymbol{\Sigma}\mathbf{V}^\top$ as the SVD of $\mathbf{X}_v\mathbf{Z}^\top$, and then we have

$$\max_{\mathbf{A}_v} \mathrm{Tr}\left(\mathbf{X}_v\mathbf{Z}^\top\mathbf{A}_v^\top\right) \Leftrightarrow \max_{\mathbf{A}_v} \mathrm{Tr}\left(\mathbf{V}^\top\mathbf{A}_v^\top\mathbf{U}\boldsymbol{\Sigma}\right). \tag{83}$$

Noting that $\mathbf{V}^\top\mathbf{A}_v^\top\mathbf{U}$ is orthogonal and that the diagonal elements of $\boldsymbol{\Sigma}$ are non-negative, we have

$$\max_{\mathbf{A}_v} \mathrm{Tr}\left(\mathbf{V}^\top\mathbf{A}_v^\top\mathbf{U}\boldsymbol{\Sigma}\right) \leq \mathrm{Tr}\left(\boldsymbol{\Sigma}\right), \tag{84}$$

where equality is attained if and only if $\mathbf{V}^\top\mathbf{A}_v^\top\mathbf{U}$ is an identity matrix. Accordingly, $\mathbf{A}_v$ is equal to $\mathbf{U}\mathbf{V}^\top$.

## H.2. Z Sub-problem

The problem concerning $\mathbf{Z}$ is

$$\min_{\mathbf{Z}} \sum_{v=1}^V \|\mathbf{X}_v - \mathbf{A}_v\mathbf{Z}\|_F^2 - \lambda_1 \mathrm{Tr}\left(\mathbf{F}^\top\mathbf{Z}^\top\mathbf{Z}\mathbf{F}\right) + \lambda_3 \sum_{v=1}^V \|\mathbf{F} - \mathbf{Z}^\top\mathbf{G}_v\|_F^2 \tag{85}$$

$$\text{s.t. } \mathbf{Z} \geq 0, \mathbf{Z}^\top\mathbf{1}_m = \mathbf{1}_n.$$

For the first term $\sum_{v=1}^V \|\mathbf{X}_v - \mathbf{A}_v\mathbf{Z}\|_F^2$, utilizing trance operation we can obtain

$$\min_{\mathbf{Z}} \sum_{v=1}^V \|\mathbf{X}_v - \mathbf{A}_v\mathbf{Z}\|_F^2 \Leftrightarrow \min_{\mathbf{Z}} \sum_{v=1}^V \mathrm{Tr}\left(\mathbf{X}_v\mathbf{X}_v^\top - 2\mathbf{X}_v^\top\mathbf{A}_v\mathbf{Z} + \mathbf{Z}^\top\mathbf{A}_v^\top\mathbf{A}_v\mathbf{Z}\right) \Leftrightarrow \min_{\mathbf{Z}} \sum_{v=1}^V \mathrm{Tr}\left(\mathbf{Z}^\top\mathbf{Z} - 2\mathbf{X}_v^\top\mathbf{A}_v\mathbf{Z}\right), \tag{86}$$

where $\mathbf{Z}^\top\mathbf{A}_v^\top\mathbf{A}_v\mathbf{Z} = \mathbf{Z}^\top\mathbf{Z}$ holds since $\mathbf{A}_v^\top\mathbf{A}_v = \mathbf{I}_m$.

For $\sum_{v=1}^V \|\mathbf{F} - \mathbf{Z}^\top\mathbf{G}_v\|_F^2$, similarly, we have

$$\min_{\mathbf{Z}} \sum_{v=1}^V \|\mathbf{F} - \mathbf{Z}^\top\mathbf{G}_v\|_F^2 \Leftrightarrow \min_{\mathbf{Z}} \sum_{v=1}^V \mathrm{Tr}\left(\mathbf{F}^\top\mathbf{F} - 2\mathbf{F}\mathbf{G}_v^\top\mathbf{Z} + \mathbf{Z}^\top\mathbf{G}_v\mathbf{G}_v^\top\mathbf{Z}\right) \Leftrightarrow \min_{\mathbf{Z}} \sum_{v=1}^V \mathrm{Tr}\left(\mathbf{Z}^\top\mathbf{G}_v\mathbf{G}_v^\top\mathbf{Z} - 2\mathbf{F}\mathbf{G}_v^\top\mathbf{Z}\right). \tag{87}$$

Minimizing the trace is equivalent to minimizing each diagonal element. Through trace element-wise minimization, we equivalently transform Eqs. (86) and (87) respectively as

$$\min_{\mathbf{Z}_{:,j}} \mathbf{Z}_{:,j}^\top V\mathbf{I}_m\mathbf{Z}_{:,j} - 2\left(\sum_{v=1}^V \mathbf{X}_v^\top\mathbf{A}_v\right)_{j,:} \mathbf{Z}_{:,j}, \tag{88}$$

and

$$\min_{\mathbf{Z}_{:,j}} \mathbf{Z}_{:,j}^\top \left(\sum_{v=1}^V \mathbf{G}_v\mathbf{G}_v^\top\right)\mathbf{Z}_{:,j} - 2\left(\mathbf{F}\sum_{v=1}^V \mathbf{G}_v^\top\right)_{j,:} \mathbf{Z}_{:,j}. \tag{89}$$

Then, for the term $\mathrm{Tr}\left(\mathbf{F}^\top\mathbf{Z}^\top\mathbf{Z}\mathbf{F}\right)$, in conjunction with **Theorem** 2, we can equivalently transform the problem $\min_{\mathbf{Z}} - \mathrm{Tr}\left(\mathbf{F}^\top\mathbf{Z}^\top\mathbf{Z}\mathbf{F}\right)$ as

$$\min_{\mathbf{Z}_{:,j}} -\mathbf{Z}_{:,j}^\top\mathbf{Z}_{:,j} - 2\sum_{a\in\mathcal{C}_i\backslash\{j\}} \mathbf{Z}_{:,a}^\top\mathbf{Z}_{:,j}. \tag{90}$$

Therefore, the problem (85) reduces to

$$\min_{\mathbf{Z}_{:,j}} \mathbf{Z}_{:,j}^\top \left((V - \lambda_1)\mathbf{I}_m + \lambda_3 \sum_{v=1}^V \mathbf{G}_v\mathbf{G}_v^\top\right)\mathbf{Z}_{:,j} - 2\left(\left(\sum_{v=1}^V \mathbf{X}_v^\top\mathbf{A}_v\right)_{j,:} + \lambda_1 \sum_{a\in\mathcal{C}_i\backslash\{j\}} \mathbf{Z}_{:,a}^\top + \lambda_3 \left(\mathbf{F}\sum_{v=1}^V \mathbf{G}_v^\top\right)_{j,:}\right)\mathbf{Z}_{:,j}$$

$$\text{s.t. } \mathbf{Z}_{:,j} \geq 0, \mathbf{Z}_{:,j}^\top\mathbf{1}_m = 1. \tag{91}$$

Combined with **Theorem** 3, we have $(V - \lambda_1)\mathbf{I}_m + \lambda_3 \sum_{v=1}^V \mathbf{G}_v\mathbf{G}_v^\top$ is a non-zero matrix. (The regularization hyper-parameter $\lambda_3$ is generally not set to zero.) As a result, Eq. (91) is a quadratic programming problem, and can be easily solved using off-the-shelf software packages.

### H.3. $\mathbf{G}_v$ Sub-problem

The problem concerning $\mathbf{G}_v$ is

$$\min_{\mathbf{G}_v} \lambda_3 \sum_{v=1}^{V} \left\| \mathbf{F} - \mathbf{Z}^\top \mathbf{G}_v \right\|_F^2 + \lambda_4 \left( \sum_{v=1}^{V} \sum_{j=1}^{k} \left[ \max \left( 0, \frac{\mathbf{1}_m^\top [\mathbf{G}_v]_{.,j}}{m} - \frac{1+\delta}{k} \right)^2 + \max \left( 0, \frac{1-\delta}{k} - \frac{\mathbf{1}_m^\top [\mathbf{G}_v]_{.,j}}{m} \right)^2 \right] \right)$$
$$\text{s.t. } \mathbf{G}_v \in \{0,1\}^{m \times k}, \mathbf{G}_v \mathbf{1}_k = \mathbf{1}_m. \tag{92}$$

In conjunction with the view independence, Eq. (92) is equivalently transformed as

$$\min_{\mathbf{G}_v} \lambda_3 \left\| \mathbf{F} - \mathbf{Z}^\top \mathbf{G}_v \right\|_F^2 + \lambda_4 \left( \sum_{j=1}^{k} \left[ \max \left( 0, \frac{\mathbf{1}_m^\top [\mathbf{G}_v]_{.,j}}{m} - \frac{1+\delta}{k} \right)^2 + \max \left( 0, \frac{1-\delta}{k} - \frac{\mathbf{1}_m^\top [\mathbf{G}_v]_{.,j}}{m} \right)^2 \right] \right) \tag{93}$$
$$\text{s.t. } \mathbf{G}_v \in \{0,1\}^{m \times k}, \mathbf{G}_v \mathbf{1}_k = \mathbf{1}_m.$$

Keeping the assignment of other anchors unchanged, we calculate the variation of the objective function when reassigning anchor $j$ from cluster $c$ to cluster $l$.

Denote the size proportion of current (anchor) cluster as $h_j$,

$$h_j = \frac{1}{m} \sum_{i=1}^{m} [\mathbf{G}_v]_{ij}, \quad j = 1, 2, \cdots, k. \tag{94}$$

When anchor $j$ is moved from cluster $c$ to cluster $l$, the new proportions are

$$h_c' = h_c - \frac{1}{m}, \quad h_l' = h_l + \frac{1}{m}. \tag{95}$$

Define

$$f(x) = \max \left( 0, x - \frac{1+\delta}{k} \right)^2 + \max \left( 0, \frac{1-\delta}{k} - x \right)^2. \tag{96}$$

Since other (anchor) clusters maintain unchanged, we have that the change in the term $\sum_{j=1}^{k} \left[ \max \left( 0, \frac{\mathbf{1}_m^\top [\mathbf{G}_v]_{.,j}}{m} - \frac{1+\delta}{k} \right)^2 + \max \left( 0, \frac{1-\delta}{k} - \frac{\mathbf{1}_m^\top [\mathbf{G}_v]_{.,j}}{m} \right)^2 \right]$ is

$$\Delta_{gvm} = f(h_c') + f(h_l') - f(h_c) - f(h_l) = f\left( h_c - \frac{1}{m} \right) - f(h_c) + f\left( h_l + \frac{1}{m} \right) - f(h_l). \tag{97}$$

Combined with Eqs. (93), (97) and **Theorem** 4, therefore, we can derive that when moving anchor $j$ from cluster $c$ to $l$, the objective change about $\mathbf{G}_v$ is

$$\Delta_{zg} = \lambda_3 \Delta_{zgf} + \lambda_4 \Delta_{gvm}. \tag{98}$$

After finding

$$l^* = \arg\min_l \Delta_{zg}, \tag{99}$$

we move anchor $j$ from cluster $c$ to cluster $l^*$, and accordingly set $[\mathbf{G}_v]_{jc} = 0$ and $[\mathbf{G}_v]_{jl^*} = 1$.

Besides, denote the term $\mathbf{F} - \mathbf{Z}^\top \mathbf{G}_v$ in Eq. (51) as $\mathbf{R}$. To prepare for the movement of the next anchor, since $\mathbf{F}$ and $\mathbf{Z}$ are given and $\mathbf{G}_v$ has been updated, we need to update $\mathbf{R}$. Specially, combined with

$$[\mathbf{G}_v]_{jc} : 1 \mapsto 0 \quad \text{and} \quad [\mathbf{G}_v]_{jl^*} : 0 \mapsto 1, \tag{100}$$

we can update $\mathbf{R}$ by

$$\mathbf{R}_{:,l^*} \leftarrow \mathbf{R}_{:,l^*} - \mathbf{Z}_{j,:}^\top \quad \text{and} \quad \mathbf{R}_{:,c} \leftarrow \mathbf{R}_{:,c} + \mathbf{Z}_{j,:}^\top. \tag{101}$$

Then, the proportions $h_{l^*}$ and $h_c$ are updated accordingly via Eq. (95).

### H.4. F Sub-problem

The problem concerning $\mathbf{F}$ is

$$\min_{\mathbf{F}} -\lambda_1 \operatorname{Tr}\left(\mathbf{F}^\top \mathbf{Z}^\top \mathbf{Z} \mathbf{F}\right) + \lambda_2 \left[\left\|\frac{\mathbf{F}^\top \mathbf{F}}{n} - \frac{\mathbf{I}_k}{k}\right\|_F^2 + \frac{1}{k}\sum_{j=1}^{k}\left(\frac{\sigma_{\mathcal{C}_j}}{\mu_{\mathcal{C}_j}} - \frac{\bar{\sigma}}{\bar{\mu}}\right)^2\right] + \lambda_3 \sum_{v=1}^{V}\left\|\mathbf{F} - \mathbf{Z}^\top \mathbf{G}_v\right\|_F^2 \tag{102}$$

$$\text{s.t. } \mathbf{F} \in \{0,1\}^{n\times k}, \mathbf{F}\mathbf{1}_k = \mathbf{1}_n.$$

Observing that the constraints are imposed on the rows of $\mathbf{F}$, we can optimize $\mathbf{F}$ row by row. Besides, the rows of $\mathbf{F}$ are one-hot vectors, and thus we can pick the cluster that maximizes the decrease in the objective function to assign to the sample.

Suppose sample $j$ currently belongs to cluster $r$; we consider moving it to cluster $t \neq r$.

Let $\mathbf{S} = \mathbf{Z}^\top \mathbf{Z}$. Then, we have $\operatorname{Tr}\left(\mathbf{F}^\top \mathbf{S} \mathbf{F}\right) = \operatorname{Tr}\left(\mathbf{F}^\top \mathbf{Z}^\top \mathbf{Z} \mathbf{F}\right)$. Combined with **Theorem 1**, we have

$$\operatorname{Tr}\left(\mathbf{F}^\top \mathbf{S} \mathbf{F}\right) = \sum_{i=1}^{k} \sum_{a \in \mathcal{C}_i} \sum_{b \in \mathcal{C}_i} S_{ab}, \tag{103}$$

which can be seen measuring the total sum of intra-cluster similarities.

For (sample) cluster $r$, after removing sample $j$, we have that the similarity lost is twice the sum of similarities between $j$ and all other samples in cluster $r$ (since $\mathbf{S}$ is symmetric), plus the subtraction of $S_{jj}$ (self-similarity). Specially, we have that the new similarity sum of cluster $r$ is

$$\sum_{a \in \mathcal{C}_r \setminus \{j\}} \sum_{b \in \mathcal{C}_r \setminus \{j\}} S_{ab}. \tag{104}$$

The previous similarity sum can be equivalently expressed as

$$\sum_{a \in \mathcal{C}_r \setminus \{j\}} \sum_{b \in \mathcal{C}_r \setminus \{j\}} S_{ab} + 2 \sum_{a \in \mathcal{C}_r \setminus \{j\}} S_{aj} + S_{jj}. \tag{105}$$

Therefore, the change in intra-cluster similarity for cluster $r$ can be characterized as

$$\Delta_r = 2 \sum_{a \in \mathcal{C}_r, a \neq j} S_{aj} + S_{jj}. \tag{106}$$

For cluster $t$, after adding sample $j$, its similarity sum becomes

$$\sum_{a \in \mathcal{C}_t} \sum_{b \in \mathcal{C}_t} S_{ab} + 2 \sum_{a \in \mathcal{C}_t} S_{aj} + S_{jj}. \tag{107}$$

(where $\mathcal{C}_t$ refers to the set of cluster members before the movement.)

Its previous similarity sum is

$$\sum_{a \in \mathcal{C}_t} \sum_{b \in \mathcal{C}_t} S_{ab}. \tag{108}$$

Therefore, for cluster $t$, the similarity change is

$$\Delta_t = 2 \sum_{a \in \mathcal{C}_t} S_{aj} + S_{jj}. \tag{109}$$

Combined with Eqs. (106) and (109), we have that after moving sample $j$ from cluster $r$ into cluster $t$, the change of $\operatorname{Tr}\left(\mathbf{F}^\top \mathbf{S} \mathbf{F}\right)$ is

$$\Delta_t - \Delta_r = 2 \sum_{a \in \mathcal{C}_t} S_{aj} - 2 \sum_{a \in \mathcal{C}_r, a \neq j} S_{aj}. \tag{110}$$

It indicates that removing sample $j$ from cluster $r$ reduces the intra-cluster similarity by $2\sum_{a\in\mathcal{C}_r, a\neq j} S_{aj}$ while adding $j$ into cluster $t$ increases the similarity by $2\sum_{a\in\mathcal{C}_t} S_{aj}$.

Therefore, for the objective $\mathrm{Tr}\left(\mathbf{F}^\top\mathbf{Z}^\top\mathbf{Z}\mathbf{F}\right)$, the change is

$$\Delta_{fz} = 2\sum_{a\in\mathcal{C}_t}\mathbf{z}_a^\top\mathbf{z}_j - 2\sum_{a\in\mathcal{C}_r, a\neq j}\mathbf{z}_a^\top\mathbf{z}_j, \tag{111}$$

where $\mathbf{z}_a$ denotes the $a$-th column of $\mathbf{Z}$.

For the term $\left\|\frac{\mathbf{F}^\top\mathbf{F}}{n} - \frac{\mathbf{I}_k}{k}\right\|_F^2$, in conjunction with the one-hot property of $\mathbf{F}$, we have

$$\mathbf{F}^\top\mathbf{F} = \mathrm{diag}\left(n_{\mathcal{C}_1}, \cdots, n_{\mathcal{C}_k}\right). \tag{112}$$

Therefore, we can obtain

$$\left\|\frac{\mathbf{F}^\top\mathbf{F}}{n} - \frac{\mathbf{I}_k}{k}\right\|_F^2 = \sum_{i=1}^k \left(\frac{n_{\mathcal{C}_i}}{n} - \frac{1}{k}\right)^2. \tag{113}$$

Accordingly, after moving sample $j$ from cluster $r$ to cluster $t$, the change in $\left\|\frac{\mathbf{F}^\top\mathbf{F}}{n} - \frac{\mathbf{I}_k}{k}\right\|_F^2$ is

$$\Delta_{si} = \left(\frac{n_{\mathcal{C}_t}+1}{n} - \frac{1}{k}\right)^2 + \left(\frac{n_{\mathcal{C}_r}-1}{n} - \frac{1}{k}\right)^2 - \left(\frac{n_{\mathcal{C}_t}}{n} - \frac{1}{k}\right)^2 - \left(\frac{n_{\mathcal{C}_r}}{n} - \frac{1}{k}\right)^2 = \frac{2\left(n_{\mathcal{C}_t} - n_{\mathcal{C}_r} + 1\right)}{n^2}. \tag{114}$$

(where $\mathcal{C}_r$ and $\mathcal{C}_t$ refer to the set of cluster members before the movement.)

For the term $\sum_{v=1}^V \left\|\mathbf{F} - \mathbf{Z}^\top\mathbf{G}_v\right\|_F^2$, via F-norm expansion, we have

$$\left\|\mathbf{F} - \mathbf{Z}^\top\mathbf{G}_v\right\|_F^2 = \mathrm{Tr}(\mathbf{F}^\top\mathbf{F}) - 2\,\mathrm{Tr}\left(\mathbf{F}^\top\mathbf{Z}^\top\mathbf{G}_v\right) + \mathrm{Tr}\left(\mathbf{G}_v^\top\mathbf{Z}\mathbf{Z}^\top\mathbf{G}_v\right). \tag{115}$$

During optimizing $\mathbf{F}$, both $\mathbf{Z}$ and $\mathbf{G}_v$ are given. Besides, $\mathrm{Tr}(\mathbf{F}^\top\mathbf{F})$ is a constant equal to $n$. Therefore, we have

$$\min_{\mathbf{F}} \sum_{v=1}^V \left\|\mathbf{F} - \mathbf{Z}^\top\mathbf{G}_v\right\|_F^2 \Leftrightarrow \min_{\mathbf{F}} -2\,\mathrm{Tr}\left(\mathbf{F}^\top\mathbf{Z}^\top\sum_{v=1}^V\mathbf{G}_v\right). \tag{116}$$

Through element-wise expansion for the trace, we have

$$\mathrm{Tr}\left(\mathbf{F}^\top\mathbf{Z}^\top\sum_{v=1}^V\mathbf{G}_v\right) = \sum_{i=1}^k\sum_{j=1}^n F_{ji}\left[\mathbf{Z}^\top\sum_{v=1}^V\mathbf{G}_v\right]_{ji}. \tag{117}$$

Combined with the 0-1 feature of $\mathbf{F}$, therefore, we have that after moving sample $j$ from cluster $r$ to cluster $t$, the change in $\sum_{v=1}^V \left\|\mathbf{F} - \mathbf{Z}^\top\mathbf{G}_v\right\|_F^2$ is

$$\Delta_{fzg} = 2\left[\mathbf{Z}^\top\sum_{v=1}^V\mathbf{G}_v\right]_{jr} - 2\left[\mathbf{Z}^\top\sum_{v=1}^V\mathbf{G}_v\right]_{jt}. \tag{118}$$

In conjunction with Eqs. (111), (114), (76) and (118), we can obtain the total change in objective loss (102),

$$\Delta_{tf} = \lambda_2\left(\Delta_{si} + \Delta_{cv}\right) + \lambda_3\Delta_{fzg} - \lambda_1\Delta_{fz}. \tag{119}$$

Then, we set $\mathbf{F}_{jr} = 0$ and $\mathbf{F}_{jt^*} = 1$ where $t^* = \arg\min_t \Delta_{tf}$.

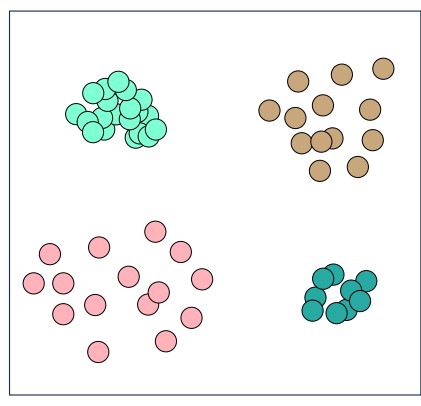 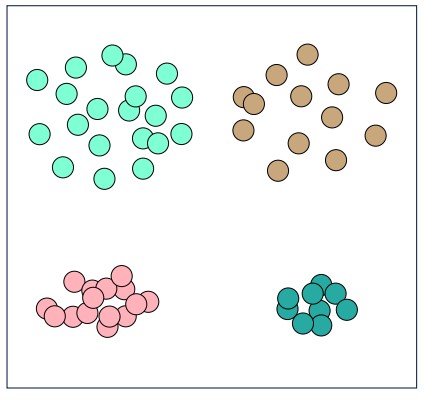 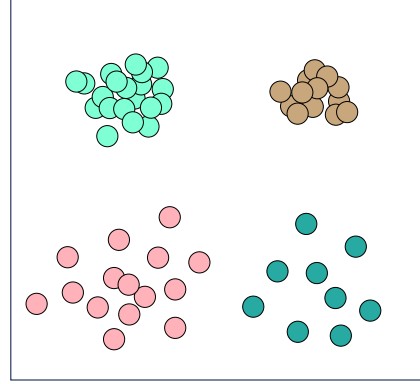

*(a)* Cluster Distribution on View 1      *(b)* Cluster Distribution on View 2      *(c)* Cluster Distribution on View 3

*Figure 7.* Multi-view Illustration of Cluster Distribution

## I. Multi-view Illustration

In multi-view scenarios, the same one cluster may exhibit distinct distributional characteristics across views. As illustrated in Fig. 7, the sizes of green cluster, pink cluster, light brown cluster, and blue cluster are 20, 15, 13, and 8, respectively. In view 1 and view 3, the green cluster is highly compact, and accordingly requires a small number of anchors to represent itself (despite its larger cluster size). In contrast, in view 2, the green cluster is relatively dispersed, and requires more anchors. Similarly, the blue cluster in view 1 and view 2 requires fewer anchors, while in view 3, it becomes more scattered and needs more anchors (despite its smaller cluster size) for adequate representation. Due to the diverse distribution characteristics, the uniform anchor allocation paradigm could fail to capture the subtle structural variations of different clusters, thereby degrading the discriminative power of anchor representations and leading to suboptimal clustering performance.

## J. Supplementary Related Work

In tackling MVC problems, besides anchor based technologies, graph, kernel, subspace, neural network methods have been widely investigated. For example, Guan et al. (2025b) learn graph structure and clustering labels jointly in a free similarity metric manner, and adjust the graph connectivity via local density to avoid overly sparse or dense connections. Wang et al. (2024) extract diverse aspects of multi-view data by picking out the most informative kernels at various scales, and utilize a dynamical partition scheme to identify the clustering across scales. Yu et al. (2025) enhance the discriminative power of subspace by imposing sparsity and low-rank constraints on the bipartite graph, and leverage a consensus mechanism to capture the underlying cluster structure consistently across views. Jin et al. (2023) employ a cross-view alignment strategy to mitigate prototype distortion, and extract supervised signals by maximizing cross-view intersection matching to regulate correspondence building. To handle streaming view data, Zhu et al. (2026b) design a mining module to determine the informative view, and update clustering assignments by dynamically calibrating view contributions. Huang et al. (2026) learn a domain discriminator to differentiate view-related distributions and a feature extractor to modulate the discriminator, and utilize adversarial agreement to maintain view invariance. Wang et al. (2026b) devise a Gaussian distribution based probabilistic framework to decrease the uncertainty in estimating missing entries, and utilize a completion module to rearrange view importance. Unlike single-consistency schemes, Jin et al. (2025) introduce a dual distribution guidance consisting of intra-view consistency and cross-view consistency to enhance the view imputation quality. Wang et al. (2026a) maximize structural pattern similarity across views through bipartite matching to build sample-level correspondences, and utilize matching weights for adaptive aggregation while maintaining view intrinsic structure. Inspired by contrastive learning attracting similar samples while repelling dissimilar ones, Guan et al. (2025a) develop a long-range component to extract global interactions and a short-range unit to attend to local regions. Zhu et al. (2026a) match broad category-level patterns with local semantics to reveal distinct semantic categories, and exploit a cross-attention strategy to extract the connections between local and global features. Under entirely unknown view-to-view correspondences, Dong et al. (2026) employ a graph matching subnetwork to establish node alignment via a consensus objective learning, and optimize label generation and alignment collaboratively to reinforce each other. Yu et al. (2023) construct a common representative-point space by permutation transformation learning to enhance semantic correspondence, and learn a shared graph across views to build unified representation. Instead of matrix-based subspace expression, Huang et al. (2024) introduce tensor-derived subspace

learning to extract view high-order representations, and make use of a semantics-maintaining regularizer to preserve cluster structure characteristics. Liang et al. (2024a) design an expectation-based learning paradigm to avoid constructing the full kernel matrix, and leverage the expectation-maximization strategy to derive optimal kernel combination weights. In addition to the above works, several other works like (Zhang et al., 2025a; Liao et al., 2025; Li et al., 2025; Zhang et al., 2025c; Ma et al., 2024; Liao et al., 2023; Chen et al., 2025b; Zhang et al., 2025b; Liu et al., 2025) have also been carefully explored.

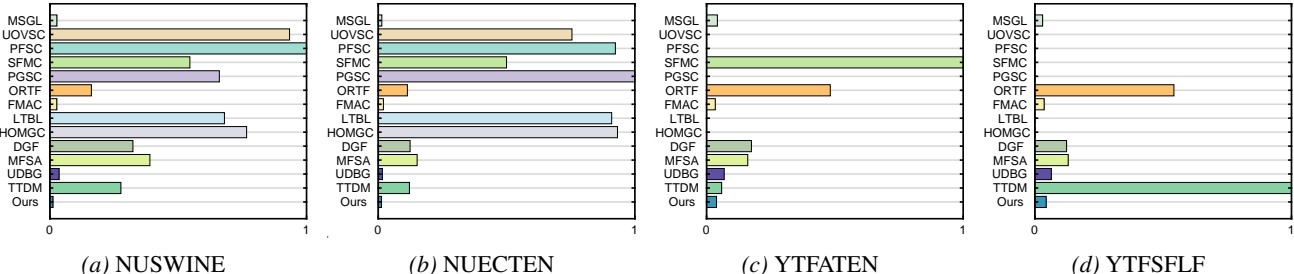

*(a)* NUSWINE      *(b)* NUECTEN      *(c)* YTFATEN      *(d)* YTFSFLF

*Figure 8.* Memory Usage Comparison on the Remaining Four Datasets

## K. An Introduction of Competitors

MSGL: This method designs a structured graph learning to progressively build affinity graphs across views, and incorporates neighbor assignment and graph regularization into subspace learning to enhance the stability.

UOVSC: This method develops a joint-optimization framework to simultaneously learn similarity matrix and cluster labels in one step, and eliminates the spectral embedding and subsequent discretization.

PFSC: This method integrates multiple partition on different views into a partition by view weight learning, and utilizes partition-level fusion to increase clustering robustness.

SFMC: This method creates a parameter-free graph fusion approach to efficiently combine multiple views, and employs anchor-based graph construction to achieve scalability.

PGSC: This method establishes pure affinity graphs without extra constraints like low-rank to learn a shared subspace representation, and utilizes graph consistency to filter out redundant noise.

ORTF: This method extracts high-order inter-view correlations via non-negative tensor factorization, and utilizes orthogonal constraints to strengthen the discriminability of learned representations.

FMAC: This method introduces second order matching relationships to rearrange anchors, and builds anchor correspondences before information fusion to guarantee cross-view consistency.

LTBL: This method presents a low-rank tensor-driven proximity learning to model within-view and cross-view relationships, and employs tensor decomposition to retain high-dimensional structural information.

HOMGC: This method devises homophily principles to refine graph structures, and leverages dynamic graph adjustment to enhance intra-cluster connectivity and clustering performance.

DGF: This method proposes a multi-graph learning to model consistent patterns and inconsistent noise, and exploits inter-view commonalities and view-specific unique properties to support comprehensive graph construction.

MFSA: This method combines feature selection and anchor graph factorization to extract informative representations, and utilizes word bag strategy to decrease feature redundancy.

UDBG: This method establishes a single optimization procedure to learn a unified anchor graph, and embeds graph generation and cluster learning to decrease the cumulative errors of separate two-stage processes.

TTDM: This method captures intricate inter-view dependencies and intra-view structural patterns via tensorized tri-factor decomposition, and leverages the high-dimensional nature of tensor data to promote cluster structure discovery.

## L. Other Memory Usage

Fig. 8 presents the memory usage on the remaining four datasets. Combined with Fig. 3, one can observe that our method still incurs relatively less memory cost, even when evaluated on large-scale datasets YTFATEN and YTFSFLF. Besides, methods MSGL and FMAC yield marginally lower memory consumption in certain scenarios. Possible reasons could be that MSGL constructs sparse graph to represent inter-sample relationships and thereby avoid the cost of maintaining multiple view-specific graphs; FMAC employs a local strategy to approximate the data manifold and accordingly circumvents pairwise similarity calculations. Despite resource-saving, they usually neglect the anchor allocation, degrading the diversity and discriminability of the final clustering representations.

Moreover, for Fig. 4, UOVSC, SFMC, FMAC and UDBG exhibit slight efficiency gains in some cases, potentially because UOVSC integrates subspace clustering pipelines into the optimization process, significantly reducing iteration overhead; SFMC adopts a lightweight matrix factorization with simple view co-regularization, avoiding the heavy refinement; FMAC adopts fixed anchor sampling and closed-form membership assignment to circumvent the need for iterative anchor optimization; and UDBG leverages precomputed view-wise measures and performs clustering via fast graph merging without solving costly global optimizations. Despite these time-saving designs, such approaches usually depend on oversimplified assumptions regarding view consistency or utilize fixed similarity fusion strategies, which may limit their ability to capture complex multi-view structures, resulting in compromised clustering results.

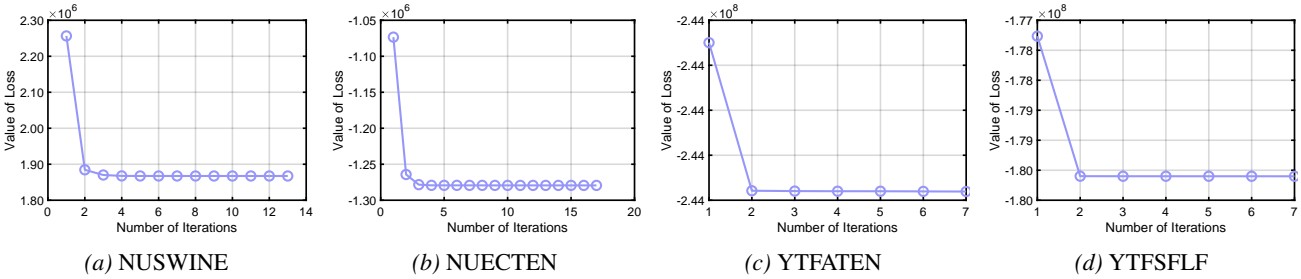

*(a)* NUSWINE      *(b)* NUECTEN      *(c)* YTFATEN      *(d)* YTFSFLF

*Figure 9.* Convergence Curve on the Remaining Four Datasets

## M. Other Convergence

Fig. 9 exhibits the loss evolution on the other four datasets. As observed, the loss is gradually decreasing and eventually reaches to a stable state, demonstrating that our method is convergent.

## N. Other Sensitivity

Fig. 10 presents the performance variation on the other four datasets. One can see that the performance exhibits minor fluctuations, which illustrates that our method is relatively robust.

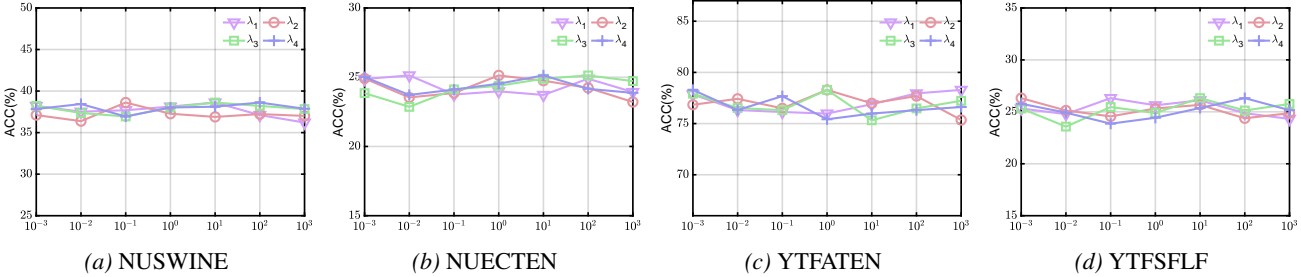

*(a)* NUSWINE      *(b)* NUECTEN      *(c)* YTFATEN      *(d)* YTFSFLF

*Figure 10.* Sensitivity Curve on the Remaining Four Datasets

