# OpenReview forum: "Fine-to-Coarse Fairness-Informed Multi-View Clustering"
_ICML.cc/2026/Conference — ICML 2026 regular_

### Official Review · Reviewer_JUrt · 2026-03-10

**Soundness:** 3
**Presentation:** 3
**Significance:** 4
**Originality:** 4
**Overall Recommendation:** 5
**Confidence:** 5

**Summary:**

This paper proposes a multi-view clustering method, FCFMVC, based on a dual-path cluster composition-aware mechanism to facilitate anchor allocation. It transfers anchor distribution supervision to sample-cluster assignment via a bipartite graph bridge, and thereby enables the model to be aware of both cluster cardinality and intra-cluster compactness during anchor learning. Further, it treats anchors as pseudo-samples and introduces per-view anchor-cluster indicator matrices with explicit constraints to regulate the proportion of anchors assigned to each cluster. Subsequently, it jointly optimizes consistency loss linking sample-clusters and anchor-clusters, creating a feedback loop to guide anchor assignment from fine-grained (anchor-level) to coarse-grained (cluster-level) structures. The linear complexity ensures scalability to large-scale scenarios.

**Compliance With Llm Reviewing Policy:**

Affirmed.

**Final Justification:**

This paper proposed a novel dual-path cluster composition-aware mechanism to help anchor allocation. This is interesting. The rebuttal addressed all my concerns, thus I keep the positive score.

**Key Questions For Authors:**

1. The paper mentions that FCFMVC may over-smooth subtle sample-anchor relationships. So, have the authors considered modifying the bipartite graph construction to mitigate this issue?
2. The variation coefficient is computed in the similarity space defined by Z. Why choosing this space? Why not the sample space?
3. In Algorithm 2 for updating F, the authors employ a block coordinate descent inspired strategy that treats cluster centroids as fixed throughout sample scanning. How many rounds of scanning are needed before convergence?

**Limitations:**

yes

**Strengths And Weaknesses:**

Strengths:
1. The proposed dual-path cluster composition-aware mechanism is novel and theoretically well-motivated.
2. The mathematical derivations are detailed, and the optimization pipeline is well-structured.
3. The experimental evaluation is thorough.

Weaknesses:
1. The method assumes complete multi-view data, and does not address incomplete scenes with missing partial views, which may limit its applicability to imperfect data environments.
2. The description of the initialization strategy is insufficient. Adding more descriptions will help clarify the implementation details.
3. The anchor-cluster indicator matrix Gv​ is optimized via row-wise scanning, which is effective but may not be the most efficient way. A greedy batch update may further reduce runtime.

---

> ### Author Rebuttal · Authors · 2026-03-31
>
> We greatly appreciate Reviewer JUrt for the thoughtful and valuable feedback on our manuscript. We have considered all the points raised, and  hope the following response addresses your concerns.
>
> **W1--Answer:**  Our primary contribution in this work is to address the anchor allocation problem in multi-view clustering scenarios. To establish a solid foundation, we first focus on the complete-view setting, which remains a widely studied and practically relevant scenario. We recognize that extending FCFMVC to handle incomplete views is an important direction. We have briefly discussed this limitation in Section 6 "Limitations and Concluding Remarks''. In the future, we will attempt to develop an extended framework suitable for incomplete scenes.
>
> **W2--Answer:**  We initialize $\mathbf{A}_v$ with orthogonal matrix,  $\mathbf{Z}$ with random one-hot column vectors, $\mathbf{F}$ with random one-hot row vectors,   $\mathbf{G}_v$ with random one-hot row vectors. Such initialization satisfies the flexible domain constraints. We will add these details in the next version.
>
> **W3--Answer:** The row-wise scanning approach for optimizing $\mathbf{G}_v$ is chosen for its simplicity and guarantee of monotonic objective decrease. Each reassignment is evaluated individually, ensuring that the loss does not increase. In contrast, batch updates may introduce oscillations or require careful tuning of batch size to maintain convergence guarantees. Besides, kindly note that during the updating of $\mathbf{G}_v$, it involves modifying the matrix $\mathbf{R}$. A greedy batch update could cause the updating of $\mathbf{R}$ to become hard.
>
> **Q1--Answer:**   The over-smoothing issue we acknowledged in Section 5.1 (e.g., on datasets CITSER and NUECTEN) arises because the bipartite graph $\mathbf{Z}$ enforces view consistency, which can potentially average out view-specific subtle structures.
>
> To mitigate this issue, a possible strategy could be that instead of forcing all views to share one $\mathbf{Z}$, we can learn view-specific bipartite graphs $\mathbf{Z}_v$ while imposing consistency and  constraints to encourage each sample to be represented by only a few anchors, preserving local structure.
>
> **Q2--Answer:**  The original sample space often has high dimensionality and contains noise. The bipartite graph $\mathbf{Z}$ maps samples into an $m$-dimensional anchor-similarity space. This acts as a denoising operation, making cluster structures more salient and the coefficient of variation more reliable.
>
> Besides, the ultimate clustering result is derived from $\mathbf{Z}$. If we computed the coefficient of variation in the original sample space, we would be regularizing a quantity that does not directly affect the clustering outcome.
>
> **Q3--Answer:**  Kindly note that the updating of $\mathbf{F}$ is performed  in the  inner-loop of Algorithm 3, and that within Algorithm 2, it  scans all samples only once. Therefore, the rounds of scanning are the same as the iteration number of outer-loop. Algorithm 2 executes a single complete scan of all $n$ samples. During this scan, each sample is considered for reassignment exactly once, with cluster centroids treated as fixed.

---

> > ### Author Rebuttal · Reviewer_JUrt · 2026-04-01
> >
> > The rebuttal addressed all my concerns.

---

### Official Review · Reviewer_CxAc · 2026-03-11

**Soundness:** 3
**Presentation:** 2
**Significance:** 3
**Originality:** 3
**Overall Recommendation:** 5
**Confidence:** 3

**Summary:**

The paper introduces a new method for multi-view clustering called FCFMVC, which aims to tackle inherent problems in anchor-learning-based MVC methods. In particular, these methods often assume a uniform distribution of anchors across clusters and implicitly assume that clusters have similar structural compactness, which may not hold in real data. The proposed method uses a dual-path cluster construction, where assignments are learned for both samples and anchors. Through this process, the model learns anchors that are more consistent with the sizes and structures of the clusters. The objective function includes constraints that encourage fairness and consistency between the cluster assignments and the learned anchors across different views. The reported results show consistent improvements over existing state-of-the-art methods.

**Compliance With Llm Reviewing Policy:**

Affirmed.

**Key Questions For Authors:**

none

**Limitations:**

yes

**Strengths And Weaknesses:**

S1. The overall formulation of the method is reasonable and technically sound.
S2. The experimental evaluation is well designed taking into account the method performance and time-memory complexity l and thorough ablation studies.
S3. The overall presentation of the paper is smooth.
S4. The method is good compared to existing MVC methods.

W1. The optimization section which feels overcrowded with remarks and theorems making it somewhat hard to follow.
W2. The paper addresses an important problem in anchor-based MVC in particular, the distribution of anchors across imbalanced clusters but does not provide concrete results on imbalanced datasets.

---

> ### Author Rebuttal · Authors · 2026-03-31
>
> We would like to express our sincere gratitude to Reviewer CxAc for the profound and perceptive  comments on our manuscript. We have carefully taken your feedback into account and hope the response below resolves your concerns.
>
>
>
> **W1--Answer:**  This arrangement was intended to rigorously support the   update rules and complexity proofs, and we will carefully revise this part in the next version.
>
> **W2--Answer:**  We would like to clarify that our experiments already include several (highly) imbalanced datasets. Specially, we summarize the number of samples in each cluster (i.e., cluster size) in the following table:
>
> | Dataset |                                                                                          Cluster Size                                                                                         |
> |:-------:|:---------------------------------------------------------------------------------------------------------------------------------------------------------------------------------------------:|
> | FORETYP |                                                                                        195, 86, 159, 83                                                                                       |
> |  CITSER |                                                                                  596, 668, 701, 249, 508, 590                                                                                 |
> | NUSWINE |                                              8,73,43,83,2,3,160,22,22,5,178,2,13,40,17,24,22,21,298,26,3,18,201,809,171,51,43,29,10,4,1166,4,524                                              |
> | NUECTEN |                                                                       497, 1422, 1249, 138, 327, 862,   243,274,288,951                                                                       |
> | YTFATEN |                                                                  6070, 5944, 4278, 4094, 3341,   3257, 3201, 2916, 2856, 2697                                                                 |
> | YTFSFLF | 2343, 1958, 2659, 2485, 2470,   3345, 2078, 3449, 3327, 2417, 2241, 3195, 2791, 2735, 2329, 2231, 2228, 3385,   2549, 2377, 2248, 2247, 2405, 1493, 2204, 2524, 2433, 2224, 1651, 2456, 27022 |
>
> Their imbalance ratio(IR, i.e., max cluster size/min cluster size) is 2.35, 2.82, 583, 10.30, 2.25 and 18.10, respectively.
>
>
> NUSWINE exhibits extreme imbalance with an IR of 583. NUECTEN and YTFSFLF exhibit high imbalance with IR values of 10.30 and 18.10, respectively. YTFATEN and FORETYP exhibit moderate imbalance.  CITSER is relatively balanced.
>
>
> Thus, for the concern, our experiments already cover datasets with severe cluster imbalance, particularly NUSWINE (IR=583) and YTFSFLF (IR=18.10).
>
>
>
> From Table 1 in the paper, we have that on NUSWINE (the most imbalanced dataset, IR=583), our method achieves the best or second-best results across all metrics, outperforming baselines like PGSC, MFSA, and UDBG. On YTFSFLF (highly imbalanced with 31 clusters), our method achieves top-2 performance and consistently outperforms strong baselines.

---

### Official Review · Reviewer_7Z2F · 2026-03-12

**Soundness:** 4
**Presentation:** 3
**Significance:** 4
**Originality:** 3
**Overall Recommendation:** 5
**Confidence:** 5

**Summary:**

Traditional anchor learning based MVC methods adopt a uniform anchor allocation strategy, which leads to inferior representation of clusters with distinct sizes and dispersion degrees. To alleviate this, this paper explicitly enables proportional anchor allocation tailored to cluster state. On the one hand, the discrete optimization pathway backpropagates cluster state to guide anchor assignment via a bipartite graph bridge. On the other hand, the direct constraint pathway leverages per-view anchor-cluster indicators to regulate anchor proportions. Then, these paths are coupled through anchor-sample label alignment, driving anchor generation from anchor-level to cluster-level structures. The model is optimized via an alternating update strategy with linear complexity. Experiments on six datasets demonstrate its effectiveness.

**Compliance With Llm Reviewing Policy:**

Affirmed.

**Final Justification:**

Thanks for the author's response. My concerns have been fully addressed, especially on orthogonality, graph bridge design, and anchor clusters. Thus, I raise the score to Accept.

**Key Questions For Authors:**

1. The method treats anchors as pseudo-samples and introduces view-specific anchor-cluster matrices. Do these matrices tend to be consistent across views, or do they capture view-specific variations in anchor-cluster assignments?

2. While orthogonality helps improve the distinguishability of anchors, could it potentially harm the performance under certain data distributions?

3. The bipartite graph is designed to be shared across all views while anchor matrices are view-specific. Why such a design?

4. The optimization pipeline includes SVD for updating Av​, quadratic programming for Z, and discrete search for Gv and F. Among these steps, which one typically dominates the computational cost?

**Limitations:**

Yes

**Strengths And Weaknesses:**

The strengths:

1. The paper identifies a fundamental and underexplored issue. By explicitly incorporating cluster cardinality and dispersion into anchor learning, it rectifies the representation of clusters.

2. The dual-path framework design is innovative, and the constructed feedback loop refines anchor allocation from multiple perspectives, enhancing the discriminative power of anchors.

3. The rigorous theoretical derivations and efficient optimization pipeline underpin its mathematical soundness, strengthening its practical applicability.

The weaknesses:

1. The framework relies on the bipartite graph Z serving as a bridge between sample and anchor spaces, but the geometric interpretation of this graph is not explored.

2. The paper only states that anchors are initialized, without specifying the manner like random sampling or k-means. Anchor learning based methods could be sensitive to initial anchor positions. A brief analysis of anchor initialization would strengthen the understanding.

3. The comparison with baselines lacks statistical significance testing. While Table 1 shows numerical improvements, it is unclear whether these gains are statistically significant across multiple runs.

---

> ### Author Rebuttal · Authors · 2026-03-31
>
> We are sincerely grateful to Reviewer 7Z2F for the valuable  and meaningful  remarks on our manuscript.  We have thoroughly considered your suggestions and hope the response below meets your expectations.
>
> **W1--Answer:**  The bipartite graph $\mathbf{Z}$ can be interpreted as a  soft assignment matrix that maps each sample to the $m$ anchors. Each view reconstructs samples as linear combinations of anchors weighted by $\mathbf{Z}$, which is shared across views. This enforces cross-view consistency, i.e., samples from different views are mapped to the same anchor-space coordinates. Besides, the matrix $\mathbf{Z}^{\top} \mathbf{Z}$ defines a similarity between samples based on their anchor association patterns. Two samples are similar if they are represented using similar anchor combinations.
>
> **W2--Answer:**  Kindly note that as shown in Algorithm 3, the anchor matrix $\mathbf{A}_v$ is first updated.  Therefore, our method is not affected by anchor initialization.
>
> **W3--Answer:**  The results reported in Table 1 are the averages across 20 runs. We will add this illustration in the next version.
>
> **Q1--Answer:**  The matrices $\\{\mathbf{G}_v\\}$ are designed to be view-specific, allowing them to capture variations in how anchors are assigned to clusters across different views. The objective function encourages consistency through the alignment term $\mathcal{L}_3$. $\mathcal{L}_3$ aligns the sample-cluster assignments $\mathbf{F}$ with the anchor-cluster assignments projected through $\mathbf{Z}^{\top}$. As a result, $\mathbf{G}_v$  is softly regularized to be consistent with the consensus clustering, while still retaining view-specific characteristics.
>
> **Q2--Answer:**  The employment of orthogonality makes anchors linearly independent, preventing redundant representations. Besides, orthogonality ensures that the SVD-based update for $\mathbf{A}_v$ has a unique closed-form solution.
>
> When views are highly correlated, enforcing orthogonality  could disrupt the inherent structure of the data.  Moreover, if the intrinsic dimensionality of the data is less than $m$, requiring $m$ orthogonal anchors compels the model to embed the data in a space of unnecessarily high dimension, potentially introducing redundancy or noise.
>
> **Q3--Answer:**  The shared $\mathbf{Z}$ forces all views to represent samples in a common anchor space, which ensures that samples are aligned and can be clustered together based on a unified representation. If $\mathbf{Z}$ was view-specific, we would have to fuse multiple graphs, potentially losing consistency. Each view has its own feature distribution and scale. The view-specific anchor matrices $\{\mathbf{A}_v\}$ allow each view to reconstruct its data using the same $\mathbf{Z}$ but with different anchor bases. Besides, the shared $\mathbf{Z}$ provides a unified clustering space. Since  the sample-cluster assignment $\mathbf{F}$ is derived from $\mathbf{Z}$, having a single $\mathbf{Z}$ yields a single clustering result. If each view had its own $\mathbf{Z}_v$, in addition to an additional fusion step, this could also introduce  complexity and potential inconsistencies.
>
>
> **Q4--Answer:**   As analyzed in Theorem 6, the updating of  $\mathbf{A} _v$, $\mathbf{Z}$, $\mathbf{G} _v$ and $\mathbf{F}$ takes  $\mathcal{O}(d _vm ^2)$, $\mathcal{O}(ndm+nkm+m ^2kV+m ^3n+mn)$,  $\mathcal{O}(nm ^2k)$,  $\mathcal{O}(mkn+mn+mkV+mnk)$,  where $d=\sum _{v=1} ^{V} d _v$. In general, the number of anchors $m$ is much  smaller than $n$. Therefore, the updating cost of $\mathbf{Z}$ is theoretically dominant. To further illustrate this, we count the optimization cost proportion of each variable. In particular, on NUECTEN, the cost proportions of  $\mathbf{A}_v$, $\mathbf{Z}$, $\mathbf{G}_v$ and $\mathbf{F}$ are 49\%, 21\%,	16\% and 14\%, respectively.  On YTFATEN, they are 57\%,	16\%, 16\%, 11\% in turn.

---

> > ### Author Rebuttal · Reviewer_7Z2F · 2026-04-05
> >
> > Thanks for the author's response. My concerns have been fully addressed, especially on orthogonality, graph bridge design, and anchor clusters. Thus, I raise the score to Accept.

---

### Official Review · Reviewer_z8CY · 2026-03-12

**Soundness:** 3
**Presentation:** 2
**Significance:** 3
**Originality:** 2
**Overall Recommendation:** 4
**Confidence:** 1

**Summary:**

The paper introduces FCFMVC, a novel method for multi-view clustering, where anchor assignment is guided in order to encourage uniform cluster representation.  Additionally, a new optimization pipeline is introduced for scalability on large scenes. Finally, the framework is valided on throuough empirical studies, against 13 baselines to demonstrate the performances of FCFMVC.

**Compliance With Llm Reviewing Policy:**

Affirmed.

**Final Justification:**

I thank the authors for their thourough and constructive answers. They have adressed my concerns about the overall lack of clarity, and provided key elements to help me understand the significance and soundness of their contribution, especially for the empirical results. Therefore, I raise my recommandation to a 4.

**Key Questions For Authors:**

- Would it be possible to device a meta-algorithm to automatically find the best hyperparameter?

- Can the authors motivate why 13 competitors are needed in the experimental section?

**Limitations:**

yes

**Strengths And Weaknesses:**

Strengths:
- The optimization pipeline allows for furhter scalability of the framework
- The extensive empirical evaluation show that the proposed framework outperforms the baselines for the proposed metrics.

Weaknesses:

- The paper is quite difficult to read through. I think the illustration presented in Appendix I should be present in the main body of the document, as it is a strong intuition guide to follow the paper, and more informative than Figure 1. Furthermore, it might be beneficial to provide further high-level intuition in Section 4, as right now, it is difficult to parse without referring to the appendix.
- The proposed objective funciton is dependant on four different hyperparameters, which lowers the practicality of the proposed framework.
- The use of 13 competitors hinders the readability and understanding of the experimental section. A narrower focus on a subset of baselines would be of great value.

---

> ### Author Rebuttal · Authors · 2026-03-31
>
> We sincerely thank Reviewer z8CY for the insightful and constructive comments on our manuscript. We have carefully considered your feedback, and truly hope the response below addresses the concerns.
>
> **W1--Answer:**   We will move this illustration to the main body of the document in the revised manuscript.
>
> For Section 4,  we will add a new introductory paragraph to provide a high-level overview of the optimization strategy before diving into the detailed update rules, for example, " The overall objective in Eq.(7) involves four sets of variables: anchor matrices  $\\{ \mathbf{A} _v\\}$, bipartite graph $\mathbf{Z}$,  anchor-cluster indicators $\{\mathbf{G}_v\}$,  and sample-cluster indicator $\mathbf{F}$.  To solve this problem,  we adopt an alternating optimization strategy, where each variable is updated while fixing the others.  Next, we present the core update rules with intuitive explanations,  and detailed derivations and proofs are provided in the appendix. "
>
>
> **W2--Answer:** As stated in Section "6. Limitations and Concluding Remarks",  we have illustrated that our method involves hyper-parameters, which somewhat limits its practicality. In the future, we will attempt to develop a non-parametric version to further enhance its practicality.
>
>
> **W3--Answer:**  We included 13 competitors so as to provide a comprehensive evaluation across diverse method families (e.g., anchor-based, graph-based, tensor-based, etc).  We categorize the 13 competitors into methodological families, as shown in the following table.
>
> |    Category    |            Methods           |
> |:--------------:|:----------------------------:|
> |  Anchor-based  | MSGL, SFMC, FMAC, MFSA, UDBG |
> |   Graph-based  |   UOVSC, PGSC, LTBL, HOMGC   |
> |  Tensor-based  |        ORTF, DGF, TTDM       |
> | Subspace-based |             PFSC             |
>
> This will help readers quickly identify which methods are most relevant to their interests. We will add these details in the next version.
>
> **Q1--Answer:**  While our current method relies on manual hyper-parameter selection with sensitivity analysis, we agree that developing an automatic hyper-parameter selection mechanism would significantly enhance practicality.
>
> For the meta-algorithm design, a possible strategy could be that we first extract dataset characteristics, such as number of samples, views, clusters, to train a predictor that provides a warm-start initialization for hyper-parameters; then, we use Bayesian optimization to efficiently refine the hyper-parameters; subsequently, we do adaptive online adjustment during optimization based on intermediate clustering quality.
>
> **Q2--Answer:**  Multi-view clustering is a diverse field with multiple methodological paradigms. In order to comprehensively evaluate the effectiveness of our proposed FCFMVC, we included representatives from major categories: anchor-based, graph-based, tensor-based, etc.  By including these 13 competitors, it provides a unified benchmark that future researchers can use for comparison. Besides, by including all 13, it ensures that our method is compared against the best possible competitor on each dataset, providing a rigorous evaluation.

---

> > ### Author Rebuttal · Reviewer_z8CY · 2026-04-01
> >
> > Thank you for the response. I greatly appreciate the clarification provided.
> >
> > Regarding W3, while it is interesting to compare to methods belonging to diverse families, it would be important to explain the need of several methods belonging to the same family of methods. For instance, in the graph-based method, why is there a need to compare to UOVSC, PGSC, LTBL and HOMGC?
> > This would help the reader better follow the experimental evaluation.

---

> > > ### Author Response · Authors · 2026-04-02
> > >
> > > We sincerely thank Reviewer z8CY for this constructive suggestion.
> > >
> > > The inclusion of several methods within the same family  is mainly to provide a diverse and thorough comparison.
> > >
> > > - Graph-based  methods learn consensus similarity graphs or partition graphs across views. While they share a common paradigm, they differ significantly in their underlying strategies.
> > >
> > > Their key strategies:
> > >
> > > UOVSC  employs unified one-step spectral clustering, and integrates graph construction and spectral partitioning into a single optimization.
> > >
> > > PGSC  adopts  partition fusion,  and combines multiple base partitions via ensemble learning.
> > >
> > > LTBL  utilizes low-rank  learning,  and models cross-view relationships using structured representations.
> > >
> > > HOMGC  introduces  homophily-enhanced graph refinement, and leverages neighborhood consistency to refine graph structures.
> > >
> > > Reason for inclusion:
> > >
> > > UOVSC represents early fusion methods that learn a unified graph directly from multi-view data.
> > >
> > > PGSC represents ensemble based graph clustering, which differs from optimization-based fusion.
> > >
> > > LTBL represents low rank based graph learning, capturing higher-order correlations beyond pairwise.
> > >
> > > HOMGC  represents graph refinement methods that post-process initial graphs to improve quality.
> > >
> > > As shown in Table 1, some graph-based methods (e.g., UOVSC) fail to run on large datasets, while our FCFMVC scales better. This demonstrates our scalability advantages relative to different graph-based strategies.
> > >
> > > Different graph-based methods excel on different datasets. For example, PGSC performs well on NUSWINE, while UOVSC performs well on FORETYP. Including all four allows us to fairly compare against the best possible graph-based competitor.
> > >
> > > - Anchor-based methods form the most relevant comparison group, sharing the anchor learning paradigm with our method.  The five baselines cover distinct strategies.
> > >
> > > Their key strategies:
> > >
> > > MSGL  performs structured graph learning, and learns a single structured anchor graph.
> > >
> > > SFMC employs scalable anchor graph fusion, and merges multiple anchor graphs adaptively.
> > >
> > > FMAC combines feature selection with anchor factorization to construct discriminative features.
> > >
> > > MFSA leverages anchor graph factorization and semi-nonnegative learning to reduce feature redundancy.
> > >
> > > UDBG   jointly optimizes anchor learning and clustering.
> > >
> > > Reason for inclusion:
> > >
> > > MSGL represents  anchor graph learning with structural regularization.
> > >
> > > SFMC represents parameter-free anchor graph fusion, a highly efficient paradigm.
> > >
> > > FMAC represents feature-aware anchor learning.
> > >
> > > MFSA represents integrated feature selection in anchor based MVC.
> > >
> > > UDBG represents unified optimization in anchor based MVC.
> > >
> > > These methods span the major innovations in anchor-based MVC: graph fusion (SFMC), feature selection (FMAC, MFSA), unified optimization (UDBG), and structural regularization (MSGL).
> > >
> > > Our method introduces fairness constraints into anchor learning. Including multiple anchor-based baselines allows us to demonstrate that the performance gains come from the fairness mechanism rather than just the anchor-based paradigm.
> > >
> > > - Tensor-based methods model cross-view relationships using higher-order tensors.
> > >
> > > Their key strategies:
> > >
> > > ORTF performs  non-negative tensor factorization, and factorizes view data into orthogonal components.
> > >
> > > DGF  models view consistency and inconsistency in a unified tensor-based objective.
> > >
> > > TTDM applies tensorized tri-factor decomposition, and captures intra-view and inter-view patterns via tensors.
> > >
> > > Reason for inclusion:
> > >
> > > ORTF represents tensor factorization methods that explicitly model view correlations with orthogonality constraints.
> > >
> > > DGF  represents  inconsistency-aware tensor learning, decomposing  views using tensor representations.
> > >
> > > TTDM represents tri-factor tensor methods, different from factorization-based methods.
> > >
> > > These three tensor methods differ in their core techniques (orthogonal factorization,consistency-inconsistency decomposition, tri-factor decomposition). Including all three provides a broad comparison against the tensor-based paradigm.
> > >
> > > Tensor methods often have high complexity. Showing that our linear-complexity method can outperform them on large-scale datasets (e.g., YTFATEN) is an important contribution.
> > >
> > > - Subspace-based methods represent another distinct paradigm.
> > >
> > > PFSC adopts partition fusion subspace learning, and combines multiple subspace partitions from different views.
> > >
> > > - In summary,  we include multiple baselines from each methodological family to provide a comprehensive evaluation. This diversity allows us to fairly evaluate FCFMVC against the best-performing strategy within each family and to demonstrate the advantages of our method.
> > >
> > > We sincerely hope this explanation clarifies the rationale behind selecting multiple methods from  each family and helps readers better understand the experimental evaluation.
> > >
> > > If any questions, please feel free to contact us.
> > >
> > > Best regards,
> > >
> > > Authors

---

### Decision · Program_Chairs · 2026-04-30

**Decision:**

Accept (regular)

**Comment:**

Reviewers consistently agree on the novelty and usefulness of the proposed dual-path cluster composition-aware design, and the rebuttal addressed most of the main technical concerns. I remain slightly cautious because some of the support comes from lower-confidence reviews, and there are still minor issues in hyperparameter dependence and implementation details. Nevertheless, these concerns do not appear to undermine the core contribution, and I therefore lean toward a weak accept.